# Mechanical Surface Treatments for Controlling Surface Integrity and Corrosion Resistance of Mg Alloy Implants: A Review

**DOI:** 10.3390/jfb14050242

**Published:** 2023-04-24

**Authors:** Vincent Santos, Mohammad Uddin, Colin Hall

**Affiliations:** 1UniSA STEM, University of South Australia, Mawson Lakes, SA 5095, Australia; 2Future Industries Institute, University of South Australia, Mawson Lakes, SA 5095, Australia

**Keywords:** surface mechanical treatment, surface integrity, corrosion resistance, biodegradable magnesium alloys, hybrid manufacturing

## Abstract

The present paper aims to provide an overview of the current state-of-the-art mechanical surface modification technologies and their response in terms of surface roughness, surface texture, and microstructural change due to cold work-hardening, affecting the surface integrity and corrosion resistance of different Mg alloys. The process mechanics of five main treatment strategies, namely, shot peening, surface mechanical attrition treatment, laser shock peening, ball burnishing, and ultrasonic nanocrystal surface modification, were discussed. The influence of the process parameters on plastic deformation and degradation characteristics was thoroughly reviewed and compared from the perspectives of surface roughness, grain modification, hardness, residual stress, and corrosion resistance over short- and long-term periods. Potential and advances in new and emerging hybrid and in-situ surface treatment strategies were comprehensively eluded and summarised. This review takes a holistic approach to identifying the fundamentals, pros, and cons of each process, thereby contributing to bridging the current gap and challenge in surface modification technology for Mg alloys. To conclude, a brief summary and future outlook resulting from the discussion were presented. The findings would offer a useful insight and guide for researchers to focus on developing new surface treatment routes to resolve surface integrity and early degradation problems for successful application of biodegradable Mg alloy implants.

## 1. Introduction

Despite its excellent biocompatibility, the limiting factor in the widespread implementation of Mg alloy in temporary support bone implants (e.g., bone plates, screws) and cardiovascular stents is its fast corrosion rate in a physiological environment. Stable metals such as stainless steel, chromium-cobalt, and titanium have been extensively used due to their durability for these short-healing period applications. Their stiffness [1] is, however, much higher than the surrounding cortical bone [2]. The difference in stiffness creates stress shielding, which leads to bone atrophy due to the lack of stimulation [3]. Alternatively, the degradable nature of Mg alloys can be capitalised on in this regard. This is because Mg’s stiffness is closer to that of the bone, reducing stress shielding, which enables controlled corrosion transfer stresses to the healing bone.

However, the inception of corrosion through the surface accelerates the undesirable and faster degradation of Mg alloys. As Mg degrades, it causes H_2_ gas generation. Human tissues around the implant site can handle a certain amount of H_2_. However, excessive H_2_ gas accumulates in tissue cavities, reducing its ability to efficiently exchange and infiltrate gases for normal biological operations. As a consequence, excessive H_2_ gas cavities cause prolonged discomfort and disturb the balance of blood cell parameters, thus decreasing survivability [4].

Researchers resorted to different strategies to address this corrosion issue. Altering the surface and substrate of Mg alloys via coating [5] or surface plastic deformation [6] has been found to retain surface integrity and improve corrosion resistance. The coating impedes the interaction between corrosive medium and substrate [7,8].

Surface integrity properties such as surface roughness (SR), cracks, impinged material, solid dissolution, and residual stress (RS) affect the corrosion performance of Mg alloys. Increased asperities increase surface area and contact with the salt solution, while impinged metallic fragments or impurities generate a galvanic cell with the surrounding α-Mg. Beneficial compressive residual stress (CRS) impedes corrosion crack propagation [9]. Moreover, the salt solution in the corrosive medium also affects the corrosion progression of the materials. For instance, the presence of Ca and P in solution produces hydroxyapatite (HA) as a corrosion product layer, thereby hindering MgCl_2_ mobility and increasing corrosion resistance.

Different coating preparation techniques, such as conversion, deposition [10], and ion implantation, and coating materials, such as polymer and ceramic [11], were used. Hybrid bioactive coatings combining PCL/HA/TiO_2_ were studied, demonstrating an effective way to control the corrosion resistance of Mg-based alloys [12]. In particular, PEO (plasma electrolytic oxidation) is found to be a reliable technique to deposit a thicker coating that is able to prevent the penetration of corrosion medium into the substrate [13]. Despite their success, the coatings still suffer from adhesion, porosity, and compactness issues, which deteriorate the intended functionality of the barrier layer in the corrosive medium [14]. Because of this challenge and complexity, researchers are still in search of better understanding coating deposition and its corrosion mechanism [15].

As an alternative to coatings, mechanical surface treatments are a viable treatment option for increasing the corrosion resistance of Mg while allowing consistent dissolution over time. Via the mechanical cold working effect, the process induces plastic straining through microstructural changes in the surface and subsurface. In the past, many studies, including machining [16], ball burnishing (BB) [17], shot peening (SP) [18], laser shock peening (LSP) [19], and surface mechanical attrition treatment (SMAT) [20], were reported to control the surface integrity and corrosion resistance of Mg alloys.

Despite overwhelming studies on mechanical surface treatment of Mg alloys, there are still discrepancies in results and conflicting findings in the literature, dictating the underlying corrosion mechanism and performance. For example, SP increases roughness significantly but results in grain refinement within the subsurface. Therefore, SPed surfaces have higher corrosion rates in the early stages, followed by slower degradation due to refined grains. LSP and SMAT show similar behaviour, but their performance can vary depending on process parameters and materials. On the other hand, BB smooths surfaces with deeper surface alteration but with very moderate grain refinement. Twinning is found to be common below the top surface of the BBed sample. As a result, BB shows higher corrosion resistance. UNSM has similar surface modification characteristics to BB. However, there is still a lack of conclusive studies dedicated to investigating the performance from the process mechanics point of view, i.e., how parameters specific to a process influence the surface modification outcome. In addition, new alloying elements (e.g., rare-earth elements) are constantly integrated to develop new Mg alloys for implant applications. The response of the current surface treatment approaches, as explained earlier, to these Mg alloys is little emphasised in the literature.

Therefore, process type, parameters, and their levels often affect the intensity and level of alternation within materials. Each technique has its own benefits as well as disadvantages from the perspective of material composition, microstructure, application, industry setup, and workpiece geometry. Thus, this paper aims to closely review and look at the recent advances in surface modification technologies and how process mechanics influences plastic deformation characteristics in terms of surface roughness, grain refinement, hardness, residual stress, and finally, the resulting corrosion behaviour for different Mg alloys. A comparison of capabilities and limitations between techniques with an outlook for prospective surface treatment solutions is presented.

## 2. Crystal Structure and Deformation Mechanism of Mg

As can be seen in Figure 1a–c, when the hexagonal close packed (HCP) crystal structure of Mg is under compressive straining, its prismatic facets can reorient at 86.3° to a basal facet while the basal to prismatic facet transition is restricted [21,22]. This increases the basal texture of the exposed surface, which coincidentally has the highest corrosion resistance in the three crystal facets of Mg [23]. However, Mg’s HCP structure prevents homogenous deformation due to the restricted slip movement in the c-axis at room temperature. Therefore, as the plastic strain increases, coarse Mg grains first generate twinning, followed by sub-grain and nanograin formations through dynamic recrystallization (DRX) (Figure 1d,e) [24]. The recrystallized nanograins have a higher basal texture, hence further improving corrosion resistance along with a more consistent and compact corrosion product layer.

## 3. Rationale for Reviewing Mechanical Surface Treatment Strategies

Since most mechanical failures are related to surface properties, surface severe plastic deformation (SSPD) has been an ideal solution to corrosion, cracking, fatigue, and wear damage problems [26,27]. The key underlying properties that can be controlled by surface SSPD are grain refinement, grain reorientation, lowered surface roughness (SR), increased surface hardness, and compressive residual stress (CRS). SSPD causes gradient microstructural change with depth up to a few hundreds of microns with nanograin refinement, followed by mild grain refinement and twinning layers. The major surface treatments that will be investigated are shot peening (SP), surface mechanical attrition treatment (SMAT), laser shock peening (LSP), ball burnishing (BB), and ultrasonic nanocrystalline surface modification (UNSM), as they have shown a great degree of potential in controlling the corrosion characteristics of Mg alloys.

To simplify the analysis, past studies with only the best-performing potentiodynamic polarisation (PDP) corrosion and its untreated counterpart are summarised in Table 1. Surface integrity includes changes in SR, microstructure, microhardness, and RS. On the other hand, corrosion performance is assessed as the change in corrosion current density and corrosion potential after surface treatment. Best performing refers to the lowest corrosion rate (i.e., the lowest corrosion current density) obtained from PDP tests and the lowest mass loss/hydrogen gas evolution rate from immersion tests. As can be seen from Table 1, the value with “Δ” only shows the relative effect after the treatment, while negative values for average R_a_ and corrosion current density (i_corr_) indicate a worse effect on surface quality and corrosion resistance, respectively. A negative value for corrosion potential (E_corr_), however, denotes an increase in E_corr_, which is desirable.

## 4. Shot Peening (SP)

In SP, a jet of metallic or ceramic shots impacts the surface to impart severe plastic deformation to the substrate (Figure 2). The indent size and the penetration depth are related to the kinetic energy of the shot media, which is dependent on the shot density, hardness, size, material, air pressure, nozzle distance, and shot angle. This can be measured through a characteristic analysis of the microstructure, microhardness, and residual stress generated within the material.

### 4.1. Surface Integrity of SP

As is shown in Figure 3a, SP increased the average SR (R_a_) by 250–520% compared to the untreated specimen. Higher roughness can be attributed to the influence of shot medium size and density, processing time, and intensity. The importance of the appropriate shot medium is highlighted in recent studies [28,29]. Smaller shot media radius and lower kinetic energy reduce roughness, but they also lower the strain penetration depth and microstructural change.

The supplied kinetic energy results in twinning, grain fragmentation, and micro- or nanograin recrystallization in the substrate, which is strongest at the surface and diminishes along the depth until it is uniform with the bulk material. The typical microstructure change with depth can be seen in Figure 3b. The coarse-grained bulk material of the untreated sample can be seen in NP (no peening). The kinetic force from the cumulative shot media impacts results in a nanograin refined section at the surface, which is above a twinning rich section of milder plastic deformation that can be seen in CSP (conventional SP). With increasing kinetic energy, either through shot velocity, treatment time, or higher mass shots, the thickness of both nanograin and twinning sections can be increased (severe SP). The thickness of the top nanograin layer is usually 40–150 μm with a mild deformation layer containing micro-grains and twinning at a depth of 150–370 μm below the top surface.

SP causes hardness improvement by 125 to 250% compared to the untreated counterpart. As shown in Figure 3c, the peak hardness is usually found at the top surface and gradually degrades linearly to match the bulk material, typically at a depth of 300–450 μm but can be as high as 1.6 mm for higher peening time and/or coverage (e.g., 4–15 min) [30]. However, the improvement in CRS and hardness can be compromised with a very rough surface due to the increased density of crack initiation sites.

In SP-treated Mg alloys, the peak CRS ranged from −50 MPa [28] to −250 MPa [30], as shown in Figure 3d. Unlike maximum hardness, which occurs at the topmost surface, maximum CRS is achieved below the top surface at as deep as 200 μm, and after the peak point, CRS decays, reaching 0 MPa at a depth of 145–550 μm. In at least two instances, the CRS curves are either very close to 0 MPa or slightly positive (i.e., tensile) on the top surface (Figure 3d). This means that tensile residual stress (TRS) is created by surface cracking during shot peening, which can negatively affect corrosion resistance. As corrosion proceeds, these initial surface cracks can propagate at a faster rate, leading to increased dissolution. On the other hand, CRS can prevent this progression by suppressing cracks, thus reducing potential corrosion sites and hydrogen production in the crack seams.

### 4.2. Corrosion Performance of SP

Figure 4a shows the relation between the corrosion potential E_corr_ and the corrosion current density i_corr_ measured from potentiodynamic (electrochemical) corrosion studies on SP-treated Mg alloys. Arrows in Figure 4a indicate the shift in corrosion resistance between the untreated and treated samples. To improve corrosion resistance, the treated surface would need to be more noble, with an increase in E_corr_ and/or a decrease in i_corr_. A higher E_corr_ will require a higher electric potential to start oxidising/decaying, and a slower reaction rate is associated with lower i_corr_ values. It can be seen that SP increases E_corr_ and i_corr_ compared to their untreated counterparts. The wide range in E_corr_ values can be attributed to the alloying elements Al, Zn, Y, and rare earth elements (REE) in AZ31, AZ91, and WE43. All of them have higher nobility, and the high grain refinement/higher grain boundary density provided by SP allow for their dissolution.

All SP-treated samples, except for high peening intensity [30] and long treatment times [18], increased i_corr_. The following analysis will separate SP’s effect on corrosion with regards to alloying elements and SP parameters. By ranking treated i_corr_ against alloying elements, WE43 and AZ91 had lower i_corr_ than AZ31 in terms of initial coarse grain size. The initial grain size is lower at 15 μm [29] and 50 μm [28] for AZ31 and 0.9 μm [30] for WE43, which suggests that as grain sizes lower and grain boundary density increases, the Cl^−^ ions have easier access to the underlying substrate, which allows for faster Mg^2+^ dissolution. The lower i_corr_ for AZ91 was due to a more thorough β-phase network that prohibited further corrosion of the underlying α-phase whereas AZ31 had an uneven network of β-phase that formed galvanic cells with the surrounding α-Mg phase [49]. The Mg_17_Al_12_ β-phase concentration on AZ91 was reduced through the SP treatment through solid dissolution, which further improved corrosion service life as seen in Figure 4c [18]. Untreated WE43’s higher i_corr_ stems from galvanic cell formation with its α-Mg and Zr β-phase while the Y β-phase had a negligible effect on i_corr_. The difference between the two SP-treated WE43 studies is the processing time: 20 s in [31] and 190 min in [30]. The shorter processing time lowered SPD, which led to less grain refinement, lower coverage, and poor solid dissolution of β-phases. While the study in [30] led to an improvement in i_corr_ post-SP treatment, it had a higher i_corr_ than the study in [31]. This could be due to higher NaCl concentration and higher R_a_ considered in [30] study.

The SP parameters of treatment time and shot diameter were compared to the resulting i_corr_, as shown in Figure 5.

Studies with no treatment time stated but referring to an almen intensity value have been given a short treatment time of 30 s based on related SP literature [51]. Due to the competing effects of the initial conditions, treatment parameters, surface integrity changes, and salt solution, identifying the optimal input value for SP treatment time and shot diameter is not easy. Arrows have been used to represent the qualitative effect of each property on i_corr_.

For treatment time, the analysis can be separated between independent research on treatment duration [28,29,48] while the WE43 alloy will be used to examine its effects across multiple studies [30,48,50]. All independent research on treatment time has shown an increase in i_corr_ compared to the untreated samples. This is due to the very low CRS gained on the immediate surface, as seen in Figure 3d [28,29]. The low-mass, 0.1-mm-diameter shots used in the AZ31 study [29] did not increase CRS significantly as the treatment time rose from 30 s to 7.5 min. The other AZ31 study [28] had 2, 7, and 14 s total treatment time with 0.3 mm diameter shots, which did not produce enough cumulative strain to significantly raise the CRS. Lastly, the WE43 study [48], which was SP treated in 60, 90, 120, and 240 s with 0.8 mm steel shots, did not measure CRS but did correlate it to corrosion resistance due to the improved crack initiation prevention and hydroxide layer stability. From independent studies, SP treatment is only viable at longer peening durations and with enough shot mass to induce a surface CRS increase to counteract the increase in surface roughness. The AZ31 study [28] did increase corrosion resistance once the top surface was electropolished, which removed asperities and exposed the higher CRS layers.

The cross-study analysis of WE43 with regards to treatment time will involve studies [30,48,50]. Both [30] and [48] were fine-grained heterogenous grains, which raised their initial i_corr_ to 208 µA/cm^2^. The 2- and 19-min treatment durations improved the corrosion resistance with the increase in CRS, which is seen in Figure 3d. With both studies using steel shots, their i_corr_ values could be much lower if treated with glass or Zr ceramic shots at higher sizes to preserve the shot mass. This is to prevent Fe impingements from forming galvanic cells with the Mg substrate. The other WE43 study was only treated for 20 s with glass shots [50] and had a lower i_corr_ compared to the previously mentioned studies but was worse than its untreated counterpart. The reasons for the worse i_corr_ for the SP sample were a very high SR with a max R_z_ measurement of 240 µm and low surface CRS based on similar CRS curves with 0.3 mm shots [28]. The slightly lower salt solution (0.5 wt% NaCl) also decreases the i_corr_ by decreasing the concentration of Cl^-^ ions to form mobile MgCl_2_.

Like the independent study on treatment time above, the surface corrosion performance of SP in general is heavily dependent on the competing effects of grain size and surface roughness against the provided CRS. When all other considerations have been removed, such as initial substrate conditions, shot composition, and salt solution, the duration of peening should be greater than 2 min for the accumulation of CRS within the substrate surface to decrease crack propagation rates and to form a denser hydroxide layer to slow Mg dissolution.

The optimum shot size is within the 0.3–0.8 mm range based on the low surface CRS attained from the 0.15 mm AZB shots used in the 7.5 min treatment of the AZ31 study [29]. If sufficient time has been given for CRS accumulation, sizes larger than 0.3 mm ceramic Zr or 0.4 mm glass are sufficient.

Long-term immersion tests highlight the advantages of the sublayer microstructural changes in terms of the formation and stability of the hydroxide film.

Peening energy and peening time affect corrosion. For example, the effectiveness of high-energy SP (HESP) for 4 min and SP for 15 min can be seen in Figure 4b, where the surface is nearly intact after exposure to salt spraying for 24 h in 0.9 wt% NaCl [30]. The 24 h salt spray test on SPed WE43 exhibited that the corrosion rate increased in the first 16 h but outperformed the untreated sample within 16–24 h. High SR may be responsible for faster degradation in the early stages. However, SP halted the propagation of cracks and had fewer instances of pitting, which slowed down the degradation as the test progressed.

## 5. Surface Mechanical Attrition Treatment (SMAT)

SMAT utilises attrition media that is enclosed with the exposed work piece, as seen in Figure 6a. The enclosure is vibrated at a fixed frequency with attrition media, typically one order of magnitude larger than SP, free to rebound and impact the exposed work piece. Unlike SP, the impact direction is random for SMAT. This can impart less normal force since the impact angle is shallower, which can result in lower depth penetration for grain refinement (Figure 6b). However, the higher mass of the attrition media and processing time can offset this limitation.

### 5.1. Surface Integrity of SMAT

As shown in Figure 7a, the SR of SMAT-treated samples increased by 230–550% compared to the untreated samples. Although both SP and SMAT have similar SR, the differences in media sizes create slightly different surface topography. As is seen from Figure 7b, SP’s smaller media form a surface with a higher density of impact craters with shallower peak-to-trough heights, while SMAT (see Figure 7c) forms wider impact craters [31,35].

As shown in Figure 7a, the 3-min SMAT treatment of extruded AZ31 at 20 Hz [34] has resulted in a very rough surface as high as R_a_ = 4.5 µm. Meanwhile, a similarly short 2-min SMAT treatment on pure annealed Mg [33] resulted in a relatively smoother surface. The difference stems from the higher hardness of extruded AZ31 compared to annealed Mg. The softer pure Mg surface deformed easier regardless of steel ball contact angle, while the harder AZ31 surface only deformed at nearly perpendicular steel ball contact angles. This results in an uneven plastic deformation distribution throughout the surface for extruded AZ31. SMATs were run for longer periods at 40 min [20] and 20 min [53], and it is found that regardless of material, longer SMAT treatment leads to lower SR due to overlapping adjacent impact craters.

Iron contaminants were identified by EDS measurements in SMAT-treated AZ31 [53,57]. This means that the process must prevent potential fragment impingement [20,32]. Microhardness, as shown in Figure 7d, for SMAT was between 1.6–3 times bulk hardness, with the peak hardness located on the top surface. In-depth hardness linearly decays until it matches the bulk at various thicknesses, depending on the initial grain size and intensity. Softer Mg alloys such as pure Mg [33] and Mg-1Ca [20,35,56] that have been annealed resulted in a hardness increase as deep as 900–1000 µm while harder Mg alloys such as AZ31 [57] and AZ91 [35,56] only reached 100–500 µm in microhardness change.

The residual stress trend from SMAT [56] is shown in Figure 7e. The CRS of SMAT matches that of SP with higher intensity but requires deeper measurements to find the penetration depth of SMAT. Deeper measurements would indicate the location of the crossover between compressive and tensile residual stress, which is usually deeper than the last instances of twinning in sample cross-sections or hardness curve shift to bulk material values. The mixture of surface cracks and CRS is likely the cause of the peak appearing 100 μm below the surface, similar to the CRS curves of SP (Figure 7e).

SMAT causes nanograin refinement due to the higher mass of the media. Even low intensity parameters, such as 2 mm media and 30–40 min treatment times, resulted in as fine as 7 nm and 71 nm grain refinement [20,35]. Like SP, the microstructure depth profile for SMAT (Figure 8) can be separated into regions of MPD that contain twinning and subgrains at the intermediate region and SPD that contain finer subgrains and DRX nanograins at the top surface region. The total MPD depths have been reported in the range of 300–1000 µm while SPD depths have been 50–300 µm from the treated surface.

### 5.2. Corrosion Performance of SMAT

A comparison of the PDP corrosion summary in terms of E_corr_ and i_corr_ for SMATed samples is shown in Figure 9a. The E_corr_ change before and after SMAT is mostly positive due to the dissolution of alloying elements and ultrafine grain sizing. Unlike SP studies that only had an improvement in i_corr_ for the highest intensity setting and longer processing time, the two SMAT studies that improved i_corr_ involved lower intensity SMAT at 2-mm media, 2- and 30-min treatment durations, and a 20 Hz frequency. Nanograin refinement is the likely reason for the improvements in AZ91 and pure Mg samples [33,35]. For instance, unlike SP, SMAT’s i_corr_ decrease is significant, with a 94% and 85% reduction in i_corr_ for AZ91 and Mg, respectively.

However, depending on the process parameters, SMAT-treated surfaces could be very chaotic, contaminated with potential iron impingement and surface crack formation. The galvanic cell formation between Fe and Mg substantially increases the corrosion rate [34,53,58] while the surface defects are prone to corrosion attack and an unstable hydroxide film [35]. To address this, extra post-processing, such as grinding, is often used to remove the impinged Fe particles [54]. This has been evidenced by lower i_corr_ and higher E_corr_ for pure Mg and Mg-1Ca samples in PDP tests [34,54].

Very limited studies reported long-term immersion results for SMATed samples. As shown in Figure 9b,c, SMAT caused higher mass loss in AZ91 [56] and a steeper hydrogen gas evolution rate in AZ31 (Figure 9b) [34], compared to the untreated sample. Low coverage was reported to be the reason for this mass-loss hike. This is quite consistent with low CRS by SMAT, as shown in Figure 7e, which might cause faster salt solution penetration and stress corrosion crack propagation through the substrate.

## 6. Laser Shock Peening (LSP)

In LSP, a laser is shot onto an ablative layer, leading to a plasma explosion that emits a high-pressure wave that propagates through the material as seen in Figure 10 [59]. LSP is reliant on the proper control of the beam divergence, pulse energy, spot diameter, laser pulse width, laser power intensity repetition rate, overlapping rate, laser wavelength, and number of laser pulse impacts [60].

### 6.1. Surface Integrity of LSP

As is shown in Figure 11a, LSP increases SR, but the increase is significantly lower, i.e., almost half, when compared with either SP or SMAT. The ablative surface explosion sites can be programmed and mapped across the surface, which lowers the surface randomness and overlap. Though coverage can be improved through proper mapping, a further increase in coverage with 75% overlap leads to a higher SR of 6.3 µm compared to the SR of 3.5 µm produced with 25% overlap [36]. Optimal coverage is thus needed to achieve the desired SR.

Hardness improvement on LSP-treated Mg alloys is shown in Figure 11b. Like SP and SMAT, the peak hardness is observed at the top surface, and hardness decays along the depth until it matches bulk hardness at a depth of about 500 µm. Higher SPD peak and penetration depths are advantageous for LSP due to the very low SR changes [36]. LSP showed a 144–167% increase in microhardness compared to the untreated sample [36,37]. This can be attributed to their higher SPD penetration depths and dislocation density compared to SP and SMAT.

The AM50 and MgCa0.8 samples were treated using laser power densities of 3 [37] and 5.1 GWcm^−2^ [36], respectively, which resulted in a penetration depth of 800 µm. While both shot and attrition media in SP and SMAT would rebound off the metal surface, the confining fluid in LSP would help restrict the high-pressure wave from propagating to the surrounding area. While the strain penetration depth was deeper for LSP, the surface average grain refinement for LSP-treated (8J) ZK60 was up to 17 µm in size (62% reduction) for the highest power density sample [39].

LSP can achieve nanograin size reduction down to 15.7 nm at a very high laser energy (10 J) [19]. There was no notable difference in the presence of β-phases between pre- and post-LSP [19,64]. In other words, solid dissolution of β-phases does not occur during LSP.

Figure 11c shows the CRS peak and penetration depth for LSPed samples. Unlike other treatments, CRS peaks appear on the top surface of the sample [36,38]. However, LSP does not appear to produce surface cracks. The reason why the wrought AZ31B had a total CRS depth of 2.5 mm (Figure 11c) [19] is due to the higher pulse energy density (PED) of 14.8 GW/cm^2^. Similar to hardness, the benefits attained from deeper, penetrating CRS curves can be rendered useless if the roughness is greatly increased.

### 6.2. Corrosion Performance of LSP

PDP corrosion results shown in Figure 12a exhibit that LSP increases corrosion resistance by exhibiting higher E_corr_ and lower i_corr_, leading to a more noble and corrosion-resistant Mg surface. The significantly lower SR, improved coverage, near nanograin refinement, and deeper penetration were the causes of the improved corrosion resistance.

A higher PED was found to increase the corrosion resistance of ZK60 [39] and MgCa0.8/X0 [36]. This is due to the increase in CRS and grain refinement. However, the opposite effect was observed for X0 samples, and this discrepancy can be explained by the wider PED range used in those studies: 1.19–2.79 GW/cm^2^ for ZK60 and 5.1–13.5 GW/cm^2^ for MgCa0.8. The higher PED used for the X0 sample has increased SR enough to counteract the benefits attained from SPD change.

The difference between increased corrosion resistances from pre- to post-treatment is highest for the lowest PED studies. For instance, the AM50 study by Jufang et al. applied PED at 3 GW/cm^2^ [37] and showed increased corrosion resistance by lowering i_corr_ from 3 to 0.5 μA/cm^2^ (83.3% reduction). On the other hand, the AM50 study by Luo et al. showed that higher PED at 22.8 GWcm^2^ [65] exhibited relatively reduced corrosion resistance improvement by lowering i_corr_ from 23.5 to 19.8 μA/cm^2^ (15.7% reduction).

PED in LSP must be chosen to achieve a good balance between SR and grain refinement that will provide enhanced corrosion resistance. Laser overlap count in LSP is another factor that affects corrosion resistance. At a PED of 22.8 GW/cm^2^, AM50 samples were LSP treated 1, 2, and 4 times [65], and the LSP-4 sample was found to show higher corrosion resistance in 3.5 wt% NaCl. This is due to the increased CRS and higher grain refinement as a result of the repeated peening effect.

Similar to PDP results, higher overlapped peening (66%) increased long-term corrosion resistance by lowering the release of H_2_ in Hank’s solution. The post-200-h corroded surfaces of the untreated and LSP treated with 66% overlap can be seen in Figure 12b [66]. Higher and deeper CRS due to overpeening prevented stress corrosion cracking (SCC) from propagating, which improved the stability of the substrate later in the immersion testing. The effect of PED on 10-day immersion of ZK60 samples’ is shown in Figure 12c [39]. The lowest mass loss was found at the lowest PED setting (1.19 GW/cm^2^). This means that higher PED increases roughness, hence accelerating pitting corrosion (Figure 12d). This observation aligns with the findings of the electrochemical corrosion test, as outlined earlier.

LSP’s E_corr_ and i_corr_ variations can also be explained by composition, SR, and grain size. The lack of either roughness or hardness data for the majority of the LSP studies prevents direct comparisons. However, pulse energy densities (PED) can be a proxy for either or both. Ranking E_corr_ in ascending order (not included), the AM and AZ series ranged from −1501 to −1350 mV, while the MgCa0.8 and ZK60 studies by Guo et al. had −650 and −1226 mV, respectively [36,39]. The lack of Al, which has an electrode potential of −1.67 V, for the ZK60 LSP sample and the higher electrode potential of Zn raise the E_corr_ for the ZK60 sample. The MgCa0.8 alloy only has Mg and Ca, which have low electrode potentials at −2870 and −2370 mV, respectively. A BB and MgCa0.8 study by Salashoor et al. measured an E_corr_ value of −1470 mV for their untreated sample [67].

The use of PED to compare LSP i_corr_ values is shown in Table 2. It should be mentioned that unlike SP and SMAT, all electrochemical data on LSP with Mg alloys showed an increase in corrosion resistance. The limited data on grain size change after LSP does confirm LSP’s fine grain refinement capability with grain sizes reduced to 0.016, 0.3, and 3 μm [19,38,64]. The trend seen in Table 2 is likely a comparison of LSP roughness data with pulse energy density since LSP has high grain refinement even at 2.18 GW/cm^2^ PED.

## 7. Ball Burnishing (BB)

BB uses consistent normal pressure to roll a metallic or ceramic ball across the surface to plastically deform the top layers (see Figure 13). The burnishing force and the ball diameter determine the amount of plastic deformation that can be imparted onto the surface, while the effect of feed rate is minimal [68].

### 7.1. Surface Integrity of BB

As shown in Figure 14a, SR in burnishing is very low due to the uninterrupted traversal of the ball with a constant ball pressure, which equally deforms the surface area. The reasons for the slight increase in roughness for MgCa0.8/X0 are high BB forces. The step over distance is wider, which prevents the adjacent pass from deforming the peaks of the previous pass. Like previous surface SPD treatments (e.g., SP, SMAT), surface grain size can be reduced to a low micrometre scale.

As shown in Figure 14b, the peak hardness of BBed AZ31B occurred at the top surface, followed by a shallow slope [69]. The relatively higher hardness from BB is due to the higher load applied, which causes finer grain refinement at the top layer. Too much force can cause deterioration of the surface and increase the corrosion rate. Salahshoor et al. demonstrated the highest corrosion resistance when X0 alloy was burnished at 400 N [40]. Figure 14c shows two CRS curves for BB [16,40]. It is on par with the rest of the CRS curves mentioned except for the HESP sample [30]. Maximum CRS occurs within the substrate, while the degree of CRS at the top surface is less prominent. This could be due to open pores and surface cracks, as reported for the X0 alloy [40]. Therefore, excessively high burnishing forces must be avoided to achieve a proper balance between increased compressive stress and top surface cracks or damage.

### 7.2. Corrosion Performance of BB

The relationship between E_corr_ and i_corr_ for BB treatments is shown in Figure 15a. The decrease in i_corr_ in all three studies shows that BB’s capability to lower surface area by compressing surface peaks and surface cracks leads to fewer instances of corrosion attack. BB can produce grain refinement up to the 1.4–2.3 µm range [17,70] which is a magnitude larger than either SP or SMAT. BBed AZ31B [6] showed that grain reorientation resulted in a peak increase for the basal plane while the prismatic planes were reduced. Uniformity in grain orientation can result in a more homogeneous hydroxide film, while a more basal-oriented surface has the highest corrosion resistance [23,40]. The likely culprits for the poor corrosion resistance of the BBed sample at higher loads are excessive surface plastic deformation forming undesired surface cracks, which are then folded back by the next adjacent ball pass and a faster feed rate.

To analyse pre- and post-BB corrosion performance, comparisons between similar alloys need to be made. The only similar alloys in the Tafel plot intercept graph are ZX41 [17] and the MgCa0.8/X0 alloy [40]. The lack of Zn in X0 should make it less noble than ZX41. With regards to i_corr_, the grain size of the ZX41 study (at 197 µm) was smaller than that for the X0 (at 500 µm). The lower grain size of the ZX41 BB sample worked well in conjunction with the lower SR to reduce the i_corr_ significantly more than the X0 sample at 79% and 20% reduction, respectively.

As seen in Figure 15b, strong pitting corrosion can be observed in the untreated sample, while the BB-treated sample only had minor degradation. Despite the treated surface showing the highest roughness, interestingly, it showed higher corrosion resistance. This is in contrast with the X0 study by Salahshoor et al. which showed the high-intensity samples reduced the corrosion resistance [40]. The higher number of burnishing passes for the ZX41 samples has increased corrosion resistance, with the second BB pass lowering surface peaks and fixing surface cracks from the previous pass. This is similar to the LSP-treated AM50 corrosion resistance increase with LSP reapplication [65]. The lower i_corr_ values for BB indicate that the combination of low SR, grain refinement, highly basal surface texture, and solid dissolution results in higher corrosion resistance.

The 7-h immersion test in a 5 wt% NaCl solution of BB-treated AZ31 can be seen in Figure 15c [6]. The accompanying 200-h immersion test of the AZ31 surfaces shows the durability of BB treatment by limiting pitting corrosion. It appeared that burnishing with cryogenic cooling slightly reduced corrosion pitting density and depth. The long-term corrosion resistance of burnished MgCa3.0/X3 alloys in 0.9 wt% NaCl was studied [16]. While burnishing at both 200 and 500 N produced similar hydrogen gas evolution trends, the lower SR from the 200 N sample performed slightly better. The 500 N sample will have a more stable trend if the test is extended due to its deeper SPD layer compared to the 200 N sample. The initial improvement in SR removes surface asperities, resulting in fewer instances of corrosion attack. The combination of a highly basal-textured surface and grain refinement allows the fast dissolution of the hydroxide film due to the increased grain boundary density.

The resulting CRS, surface basal texture, and grain refinement improvements by BB can increase the survivability of Mg alloys in physiological environments. The difficulty of applying BB in very narrow spaces of complex geometry, such as concave corners, limits BB’s industrial application. The preferred parameter combinations that resulted in higher corrosion resistance were mid-to-high BB force, smaller step-over distance, low feed rate, and more than one number of passes.

The electrochemical data for BB is rearranged in Table 3 to show the relationship between composition, roughness, grain size, and the electrochemical data for the BB studies. The reason for BB’s precise Tafel plot intercept positioning in Figure 15c is the limited grain size reduction (1.4 to 17.4 μm) while consistently generating very low roughness at 0.13–0.9 μm. For BB, lower SR is preferred to reduce i_corr_, which is like LSP. Both the AZ31 study by Pu et al. and the MgCa0.8 study by Salahshoor et al. follow the expected E_corr_ trend, with the Ca-containing alloy having a lower E_corr_ value [6,40]. The Ramesh et al. E_corr_ value for BB-treated ZX41 is suspiciously low due to the increased Zn concentration but is not unreasonable [17].

## 8. Ultrasonic Nanocrystal Surface Modification (UNSM)

UNSM, often named ultrasonic impact peening (UIP), has similarities to BB; however, its tool tip is not revolving relative to the rest of the tool. An attached piezoelectric tool imparts vertical ultrasonic vibration while the tool tip is moved under static loading, as seen in Figure 16 [71]. The parameters that affect the surface integrity are static load, frequency, amplitude, feed rate, step-over distance, and tool tip diameter. Unlike BB, UNSM is not continuous and can produce gaps if the feed rate is too high for the given tool’s operating frequency and amplitude.

### 8.1. Surface Integrity of UNSM

SR changes for UNSM-treated Mg are shown in Figure 17a. The addition of a vibratory motion produces overlapping dimples, making UNSM not continuous. Lower SR by UNSM can be achieved with lower feed rates and shorter stepover distances. The decrease in roughness for the LZ91 sample is due to the lower load of 1.4 N [45], which is less disruptive than the 5, 20, and 85 N loads reported in other studies [42,43,44].

The hardness of UNSM-treated Mg is shown in Figure 17b. UNSM-treated AZ91 showed a very high hardness increase from 215 HV (bulk) to 295 HV [72]. With variations on the static loading of 5 N [42], 20 N [44], and 85 N [43], the plastic deformation is steadily raised. This results in the formation of twins in AZ31B, as shown in Figure 17c (as indicated by the red arrows in the figure). The formation of twinning can also be measured through the in-depth hardness in Figure 17d, where it has a peak of 95 HV and is decaying to the bulk material hardness at 500 µm. At higher static forces, there is grain refinement at the surface, as seen in Figure 17e,f, which decreases with depth when 20 N is applied to the ZX11 sample [44].

### 8.2. Corrosion Performance of UNSM

Tafel plot intersects of UNSMed samples are shown in Figure 18a. Most studies showed that SPD induced by UNSM led to an increased E_corr_. The link between UNSM’s applied load and i_corr_ is not always proportional to the static load. For instance, the poor corrosion resistance of UNSMed AZ31 was due to the high load of 85 N, which produced surface crystal defects and a higher SR [43]. On the other hand, Hou et al. reported marginal mass loss for AZ31 in 24 h of immersion when UNSMed was at a load of 5 N (Figure 18b) [42].

Baek et al. [44] reported the lowest SR and high grain refinement on UNSMEd ZX11 at 20 N, which produced low i_corr_ by reducing surface cracks and pitting (see Figure 18c). Thus, the UNSM static load for Mg requires optimisation as low loads can lead to no grain refinement and much larger static loads can overwork harden the substrate, leading to poor corrosion resistance. The combination of proper treatment path planning leads to better coverage and ultrafine surface grain structure.

The reordered electrochemical UNSM data can be seen in Table 4. The E_corr_ trend follows the expected ranking for the UNSM-treated samples. The electrode potentials of the main alloying elements are −3040 mV for Li, −2870 mV for Ca, −1670 mV for Al, and −760 mV for Zn. The E_corr_ of the LZ91 is suspiciously low for the 9 wt% Li. The post-UNSM i_corr_ and SR trend mirrors that of the SP and SMAT, where higher post-SP/SMAT SR resulted in lower i_corr_. This is the opposite of the LSP and BB i_corr_ and SR trends, where higher SR (or its proxy) would result in higher i_corr_ values. There is no trend in the used UNSM load with i_corr_ or SR measurements. Having more information on feed rate and step-over distance would help in deciphering UNSM’s effect on surface chemistry.

## 9. Comparative Analysis of Surface Treatments

### 9.1. Surface Integrity

The average SR measurement for each treatment is shown in Figure 19a. Both SP and SMAT increase the untreated samples’ roughness up to 8–10 μm compared to LSP, BB, and UNSM, which only increase SR by fractions of microns. This is due to the displacement of the substrate with each random media impact that overlaps. LSP, BB, and UNSM, on the other hand, are controlled processes that have limited disturbances in surface topography.

According to a comparison of the relative changes in hardness shown in Figure 19b, SP and SMAT have the highest peaks that linearly degrade to the bulk material hardness between 400–500 μm. Higher peening intensities can achieve a higher peak hardness and penetration depth, such as a combination of 4 min of SP with steel balls at 700 kPa and 15 min of SP with glass beads at 400 kPa on WE43 [30]. The same effect can be achieved with a lower bulk hardness with 40 min of SMAT using 2 mm zirconium balls [20]. LSP, BB, and UNSM have deeper penetration depths at lower peak surface hardness.

The likely explanation for both trends is the higher grain refinement at the top surface with both SP and SMAT. The repeated impacts of SP and SMAT have cumulatively deformed the surface, which produces 32 nm [30], 37 nm [33], and 42 nm [54] nanograins. In contrast, LSP, BB, and UNSM can reach the 100 nm grain range at the surface with a deeper strain penetration depth [44,74].

The RS curves for each treatment are shown in Figure 19c. Unfortunately, there was no RS data for UNSM. Both SP and SMAT surface RS [27,56] are lower than the LSP and BB [38,40], which is likely due to the formation of surface cracking from SP and SMAT created from high velocity impacts with overlapping peaks that have tensile RS. The peak CRS is usually found some depth into the substrate for both SP and SMAT. The lower CRS surface peaks are likely the cause of SP and SMAT’s lower corrosion resistance in most SP and SMAT electrochemical and immersion corrosion tests.

The surface integrity comparison of the five surface mechanical treatments has shown that SP and SMAT’s chaotic nature results in significantly higher surface topography than LSP, BB, and UNSM. Nanograin refinement is attainable for SP, SMAT, LSP, and UNSM, but BB can only reach low micrograin refinement. SP and SMAT’s high grain refinement results in steeper microhardness slopes, while LSP, BB, and UNSM have lower peak microhardness with a shallower trend that can last up to 1 mm into the substrate. Finally, the surface CRS of SP and SMAT is lower than that of LSP and BB, potentially due to surface cracks from SP and SMAT.

### 9.2. Corrosion Performance

Figure 20 shows all the Tafel plot intercepts shown thus far. As previously mentioned, the highest E_corr_ and lowest i_corr_ (bottom right of the graph) were the desired properties for the highest corrosion resistance for the Mg substrate. LSP, BB, and UNSM’s i_corr_ treated intercept precision is higher than either SP or SMAT. However, the lowest i_corr_ treatments are SP, SMAT, and LSP.

The main properties to rationalise each treatment’s overall position, as shown in Figure 20, are composition, SR, grain refinement, and electrochemical testing conditions. The broad range of SP intercept values in Figure 20 can be attributed to composition for E_corr_ values and SR and grain refinement for i_corr_. Most of the SP studies used AZ and WE series Mg alloys. If the SP E_corr_ was ranked in ascending order, the resulting order in Mg alloys from least to most noble would be AZ31, then AZ91, and WE43. The studies that had higher R_a_ surprisingly had low i_corr_.

Comparisons for SMAT studies did not yield any useful connections. SP is simpler and utilises uni-directional balls that have very similar velocities at impact. Due to SMAT’s increased randomness, varying contact angles, equipment differences, and media-to-media collisions, the resulting SPD and coverage are not consistent.

A simplified version of Figure 20 is shown in Figure 21, which only shows the lowest E_corr_/i_corr_ intercept per treatment. The elemental components are AZ91 for SP (■) [18], AZ91 for SMAT (♦) [35], AM50 for LSP (▲) [37], AZ31 for BB (✕) [6], and ZX11 UNSM (●) [44].

The BB-AZ31 intercept can be explained by its low grain refinement (1.4 μm) and use of a 5 wt% NaCl salt solution, which provides no protections such as a calcium phosphate-deposited layer from either a SBF or Hank’s solution [6]. The resulting SR was improved, hence the lowering of the i_corr_ value. The SP-AZ91, SMAT-AZ91, and LSP-AM50 samples all used 3.5 wt% NaCl salt solutions and had high surface grain refinement at 0.13, 0.007, and 0.3 μm, respectively [18,35,37]. The SR measurements were not given for the SP study, but the R_a_ for the SMAT and LSP samples were 4 and 0.65 μm. The SMAT-AZ91 sample’s high SR should have resulted in a higher i_corr_ value. The very low surface grain size likely counteracted the high SR, which further lowered the i_corr_ value.

The UNSM-ZX11 study by Baek et al. should have the best performing i_corr_ given that it has phenomenal grain refinement at 0.2 μm, roughness is very low at 0.11 μm and it was submerged in SBF at 37 °C, which provides Ca and P for hydroxyapatite formation, while the other studies used differing NaCl concentrations (% wt). The other studies that used AZ31 increased i_corr_ after UNSM.

The treated Tafel plot intercepts have shown that LSP, BB, and UNSM have higher precision than SP and SMAT. The variation in SR and grain refinement for SP and SMAT increases their E_corr_ and i_corr_ more than LSP, BB, and UNSM. Surprisingly, the lowest i_corr_ measured for surface mechanically treated Mg alloys were measured for SP, SMAT, and LSP. The plot shows the importance of using the appropriate parameters for SP and SMAT, such as longer treatment times and higher SPD for SP and lower intensity and amplitude for SMAT. Meanwhile, LSP can produce improved surface chemistry at a wide range of PED, overlap percentage, and coverage numbers. Ultimately, low SR alone does not further lower corrosion resistance if it’s not simultaneously delivered with nanograin refinement.

By excluding the untreated portion of each study, some relationships between the treatment parameters and/or the resulting surface integrity can be related to the measured corrosion resistance. The treated E_corr_ was mainly connected to the main alloying elements’ electrode potential, with some outliers like the LSP MgCa0.8 study by Guo et al. and the LZ91 study by Wang et al. [36,45]. SP’s, SMAT’s, and UNSM’s increases in SR resulted in inversely proportional decreases in measured i_corr_ values, while the inverse was seen for LSP and BB. When used as proxies, high intensity or microhardness can show the relation with grain refinement and i_corr_. SP, SMAT, and BB’s increasing hardness (∝ grain reduction) correlated to lower i_corr_. LSP, on the other hand, already creates nanograin refinement at low PED intensity, which only led to increasing intensities and increasing SR, which increased i_corr_.

Based on electrochemical measurements done for pure Mg, AZ31, and AZ91 [49], the study argues that the higher concentration of the β phase (Mg_17_Al_12_) for AZ91 provides a sheet of β-phase that is more corrosion resistant than the α-β phase galvanic cell formed by the AZ31. With further exposure, the gaps in the AZ31 β-phase network allow for the galvanic cell formation of the now exposed α-phase grains underneath the uncorroded β-phase.

## 10. Hybrid Surface Treatments

Due to the complexities of Mg alloy corrosion, the use of one surface treatment routine may not provide sufficient service life in physiological environments. Thus, researchers investigated the combined treatments. For SP and SMAT, the industrial practise for improving corrosion and fatigue service life is to polish or grind the asperities and/or contaminants off the immediate surface while maintaining a suitably thick modified surface layer. For instance, electropolishing (EP) has decreased the corrosion rate of high-intensity SP-treated Mg samples (0.8 mmN) from 5.72 mm/year to 0.45 mm/year, as seen in Figure 22a [28]. SMAT-treated Mg-1Ca/X1 that previously performed poorly in PDP tests but by grinding off a 100 µm layer further resulted in a 27% reduction in i_corr_ [20]. A similar finding was reported for SMAT treated AZ31 samples by selectively removing the top rough surface layer in the range of 40–100 µm, which is demonstrated by lower i_corr_ and higher E_corr_, as shown in Figure 22b [57]. This also helped increase the long-term corrosion resistance of the SMATed surface by reducing hydrogen gas evolution and mass loss rates in the immersion test (Figure 22e).

LSP on cold-sprayed Al onto the LA43M substrate resulted in a 500% increase in corrosion resistance [75]. Coatings have been deposited on the treated surface. For instance, LSP followed by phosphate conversion coating (PCC) on AZ31 was successful in reducing i_corr_ from 3.52% to 3.01% (Figure 22c) [64]. Similarly, LSP + MAO (micro arc oxidation) coating increased the corrosion resistance by two orders of magnitude [38]. As shown in Figure 22d, synergistic BB and hydroxyapatite (HA) on AZ31 reduced i_corr_ by 16% and increased the nobility by 500 mV [76,77].

Thus, it appears that the combination of two or more treatments can yield greater corrosion resistance but requires a proper synergy between one treatment’s disadvantages and the other’s advantages. The complementing attributes of both treatments will ensure the underlying substrate will have a much better service life than relying only on one treatment or the other. Surface features such as surface roughness, surface energy, and contact angle generated by initial surface treatment are also critical to depositing additional compact and quality coatings. This has been further emphasised by Mhaede et al. in which they concluded that SP at low intensity followed by dicalcium phosphate dehydrate (DCPD) coating deposition provides a good balance of high strength and good corrosion resistance in AZ31 alloys [5]. The PEO coating on SPed AZ31 was studied, demonstrating an improvement in corrosion resistance [78].

**Figure 22 jfb-14-00242-f022:**
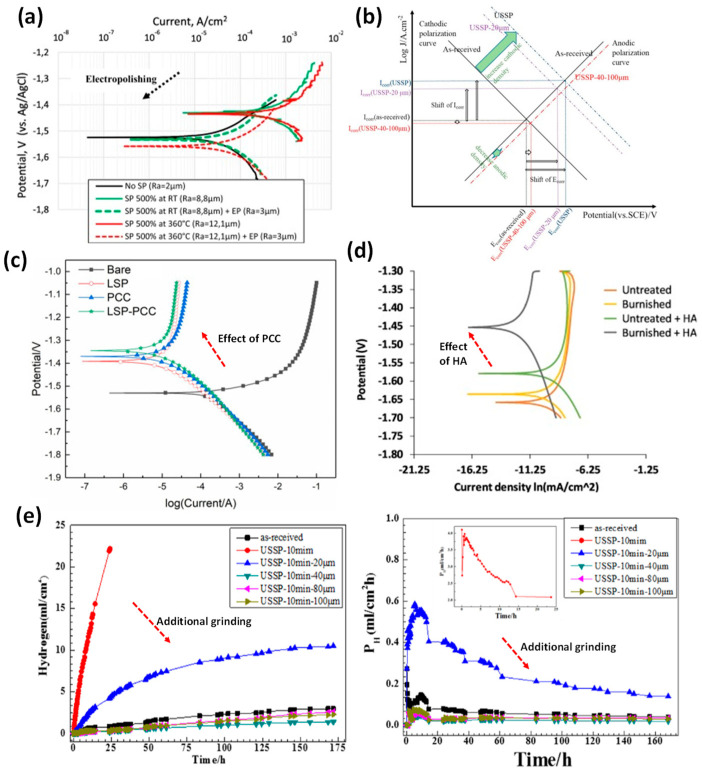
Tafel plot intercept changes pre and post treatment of Mg alloys: (**a**) SP+EP/AZ31 [28], (**b**) SMAT+EP/AZ31 [57], (**c**) LSP+PCC/AZ31 [64], (**d**) BB+HA/AZ31 [77], and (**e**) SMAT+EP/AZ31 [57].

## 11. Emerging Surface Treatment Techniques

Researchers have continuously attempted to develop new surface treatment technologies by expanding and leveraging the strengths of existing techniques to improve the surface integrity and corrosion resistance of Mg alloys. A cavitation peening is employed to increase the surface hardness and residual stress of AZ80A and AZ31B alloys [79,80]. In cavitation peening (Figure 23a), the growth and collapse of vaporous or gaseous cavities due to local pressure drop and recovery causes a peening effect on the surface, inducing local plastic deformation. Like SP or LSP, cavitation peening causes high roughness due to the local micro-peening effect (Figure 23b), but increases hardness by up to 20–40% (Figure 23c) and residual stress by up to −220 MPa (Figure 23d) for AZ80A by inducing twinning in the microstructure [80].

By introducing the ultrasonic effect in cavitation peening, hardness has increased by up to 48% with higher grain refinement of as little as 10 µm in AZ31B alloy [79]. The importance of optimising cavitation peening process parameters to achieve the desired outcome for AZ31 alloys was emphasised [81]. Because of the setup complexity and limitations of the workpiece geometry to be treated, the industrial application of the cavitation process still remains a challenge.

Very recently, Zhu et al. (2021) studied burnishing treatment followed by aluminium alloying via thermal diffusion on a pure Mg surface and demonstrated the generation of a thicker and more uniform Al-enriched surface layer that could be spontaneously passivated like an Al alloy and thus significantly enhanced the corrosion resistance (Figure 24a) [82]. It was shown that burnishing-induced grain refinement and active surface energy facilitated the formation of Mg-Al-rich precipitants (Figure 24b) that are highly insusceptible to the most stubborn galvanic corrosion (Figure 24c,d).

This surface treatment technique seems to be an effective method for industrial corrosion protection of lightweight structures made of Mg alloys, but its application in the biomedical industry as implants must be evaluated through an in vitro cell viability study. It was reported that higher aluminium content in implants causes neurological disorders or related diseases in patients. Thus, a new surface treatment strategy might need to be explored to address emerging Mg alloys with no or little aluminium content.

## 12. Surface Treatment Challenges in Additively Manufactured Mg Alloys

Due to its ability to make complex and customised geometry in a relatively faster time period, additive manufacturing (AM) is increasingly used to fabricate medical devices, including metallic and plastic implants and scaffolds. AM parts are often rough and porous, leading to undesired residual stress and cracks that compromise mechanical and fatigue properties. Mechanical surface SPD treatments such as burnishing and shot peening were used to modify the SR, grain structure, and CRS, leading to an improvement in the surface integrity and corrosion resistance of the newly 3D printed part [83].

For example, in order to leverage the benefit of the surface treatment, very recently researchers attempted in-situ roller burnishing treatment on direct energy deposition (DED) biocompatible alloys [84]. In this case, a burnishing tool attached next to the laser head rolls a hot layer of deposited material in-situ to refine grains and induce compressive stress before the second layer is deposited, and the process continues until the final product is printed (Figure 25a). Results showed that by applying in-situ roller burnishing on DED titanium alloys, favourable grain refinement, texture shape, and orientation can be achieved, which ensure homogeneity and isotropy, hence increasing tensile properties including strength by 20% and elongation by 17% (Figure 25b).

In-situ surface treatment seems very effective in industrial applications for high-melting-point hard materials, requiring less force and energy. However, the practical integration of such a technique for relatively softer materials such as Mg and its alloys is still a matter of investigation. Moreover, material microstructure, properties, and the underlying benefit resulting from surface modification must be studied before considering the use of this in-situ hot treatment.

The current challenges associated with Amed Mg alloys for biomedical applications are elucidated in recent articles [85,86]. In laser bed fusion of Mg alloys, factors such as powder characteristics, laser power density, layer thickness, and bed temperature affect the final surface texture, porosity, and internal stress. Similar issues around AMed Mg alloys are facing other AM processes such as WAAM (wire arc additive manufacturing) of AZ31 [87], paste extrusion deposition of MgP (magnesium phosphate)-based scaffolds [88], friction stir additive manufacturing of WE43 [89], and binder jetting of MgP [90]. Therefore, it is clear that surface treatment as postprocessing is inevitable to address these inherent surface integrity issues for AMed Mg alloys. Based on the analysis of all plasticity surface treatments explained earlier, ball burnishing (BB) would be a suitable method by appropriately controlling its process parameters to smooth the surface while imparting a low to mild cold working effect.

Moreover, as an alternative to high-force SPD treatment, due to the delicate structure of the Mg prints, such as tailored scaffolds, bone plates, and screws, low to moderate surface modifications such as sandblasting [91] and preheat treatment [92] can often be employed to reduce surface roughness and porosity and increase compactness and mechanical properties. A suitable coating such as HA would be used as well to augment the corrosion resistance of the AMed Mg alloy scaffold.

It seems that severe mechanical surface treatments like SP, LSP, and BB on AMed Mg alloys are little studied in the literature, which lacks holistic insights. Therefore, it poses a challenge for researchers to discover new and emerging surface treatment technology to control the corrosion resistance of biodegradable Mg alloys, and this should be the focus of future research on surface treatment processes for biodegradable Mg alloys.

## 13. Summary and Outlook

### 13.1. Summary

Controlled surface plastic deformation via mechanical surface treatment techniques is an alternative plausible way to regulate the surface integrity and corrosion resistance of Mg alloys. Regardless of the treatment technique employed, the combined influence of surface roughness, grain refinement, compressive residual stress, and undesired surface cracks induced in the material plays a crucial role in augmenting the surface integrity and thereafter degradation characteristics of Mg alloys. Interestingly, the alloy composition ranging from pure Mg, AZ31, to rare-earth WE43 and the concentration/type of saline medium contribute to the local and global corrosion of the treated surfaces.

Because of the controlled nature of plasticity deformation, LSP, BB, and UNSM increase surface finish. This is, however, not the case for SP and SMAT because of their process randomness and repeated indents by peening media, resulting in high roughness and potential surface cracks.

All treatment techniques are able to cause grain refinement within the sub-surface at a certain depth, but the density and level of refinement are highly dependent on the key processing parameters, including deformation intensity, media, overlap, time, pressure, and load. Predominantly, SP, SMAT, and LSP outperform UNSM and BB in terms of deeper and finer grain refinement.

In terms of magnitude and gradient, SMAT shows the highest microhardness, followed by SP, LSP, UNSM, and BB, in descending order. However, the compressive residual stress and depth induced by SP and SMAT are lower than that resulted from either LSP or BB due to the undesired high roughness and surface defects caused by the severe nature of plastic deformation.

Electrochemical corrosion results showed that LSP and BB-treated surfaces were more corrosion-resistant (i.e., lower i_corr_ and higher E_corr_) compared to other techniques. The lower surface roughness, moderate nanograin refinement, and lack of surface defects are responsible for the increased corrosion resistance. A similar trend was noted for long-term immersion in terms of H_2_ generation, mass loss, and pitting growth.

### 13.2. Outlook

Each surface treatment has its own unique advantage and process limitation, but the final outcome can be optimised by carefully controlling its process-specific key parameters. For instance, higher peening intensity and lower pulse energy density for SP and LSP, respectively, dictate the resulting corrosion resistance, while increased coverage, low static load, and moderate burnishing pressure for SMAT, UNSM, and BB, respectively, dictate the resulting corrosion resistance.

Since a single treatment may not be a complete solution, additional post-processing, e.g., grinding, polishing, and coating, after major surface treatment appears to be an effective means to alleviate the issue with surface defects and contamination, especially for SP and SMAT, leading to enhanced corrosion resistance. However, this is achieved at the sacrifice of additional costs and resources to be incurred on the manufacturing process line.

### 13.3. Future Research Directions

A more simple, robust, and environmentally friendly process such as ultrasonic water peening has been found to increase the surface integrity and corrosion resistance of Mg alloys. A highly corrosion-resistant Al-rich layer on a burnished Mg substrate via thermal diffusion could be another possibility, but the concentration of Al must be carefully studied and restricted to pass cell viability criteria while providing adequate surface integrity and degradation kinetics.

Another challenge is that Amed Mg alloys are not readily usable, hence often requiring a surface treatment to meet the requirements. In-situ hot laser interlayer surface treatment could resolve the issue, but this is a relatively new area of research that might be the focus of future research in surface treatment technology for Mg alloys.

It is evident that the outcome of surface treatments studied is intrinsically related to the interaction of treatment time, media, and intensity (force/pressure). Thus, an optimisation approach would be used to determine the effect of the main parameters. This will form the basis for building a predictive model for further surface treatment optimisation for new Mg alloys.

The corrosion of the surface-treated Mg alloys is dependent on the salt media, concentration, temperature, and environment. Thus, the interaction of these corrosion media with microstructure (i.e., grain refinement or modification) and phases of the materials underpinning the local and global corrosion mechanisms in the short and long terms needs to be further studied.

## Figures and Tables

**Figure 1 jfb-14-00242-f001:**
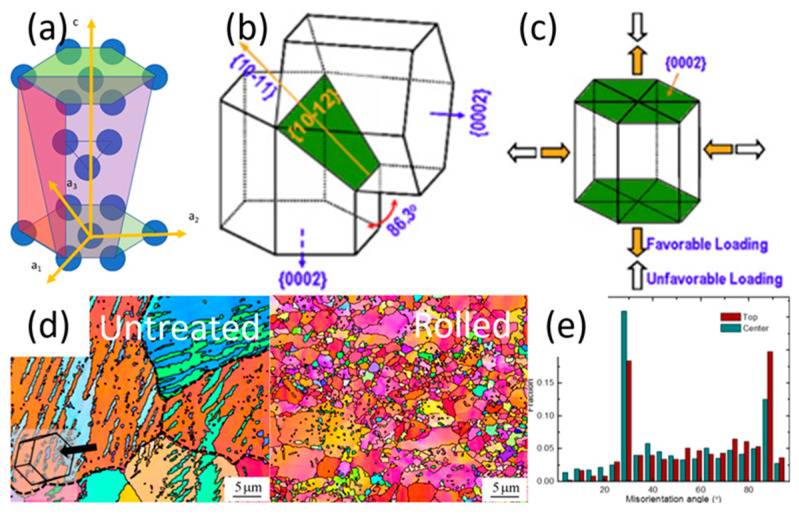
(**a**) Hexagonal–close packed crystal (HCP) structure with basal facets coloured green, prismatic facets coloured red, and pyramidal facets coloured purple. Mg tensile twinning system with (**b**) the 86.3° reorientation of the parent Mg crystal lattice (**c**) Yellow arrows indicate favour loading for tensile twinning, while clear arrows indicate no twinning for the crystal lattice; (**d**) dynamic recrystallized nanograins after rolling treatment, and (**e**) nanograin distribution [25].

**Figure 2 jfb-14-00242-f002:**
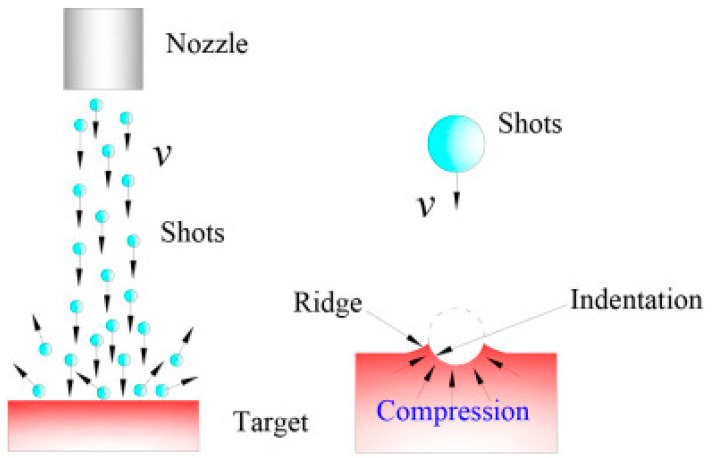
Illustration of the shot-peening schematic [46].

**Figure 3 jfb-14-00242-f003:**
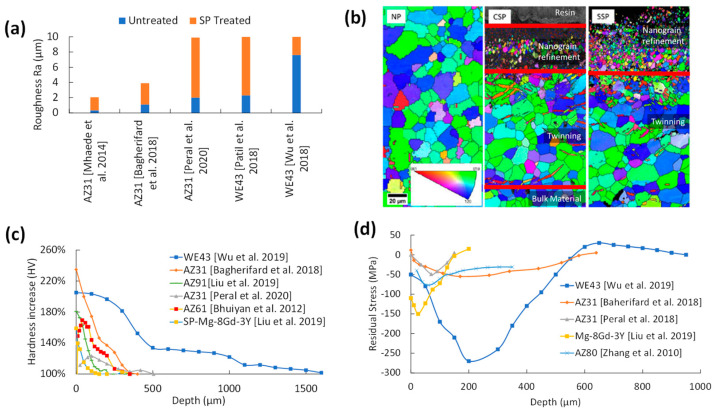
Effect of SP on (**a**) surface roughness [5,28,29,30,31]; (**b**) EBSD microstructural grain refinement [29]; (**c**) hardness [9,18,28,29,30,47]; and (**d**) residual stress [27,28,29,30,47] of Mg alloys.

**Figure 4 jfb-14-00242-f004:**
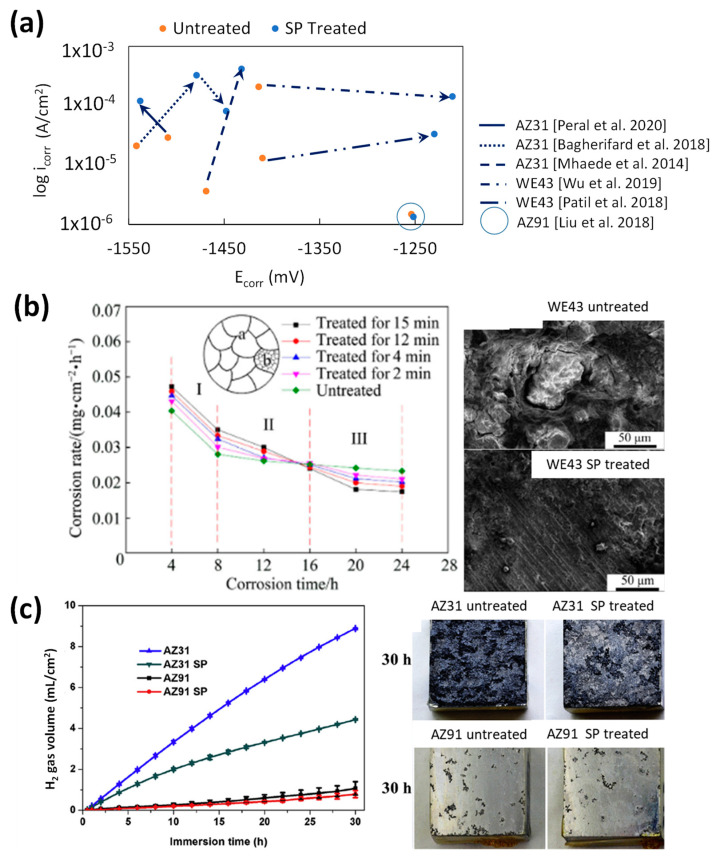
Effect of SP on (**a**) potentiodynamic corrosion [5,28,29,30,31,48], (**b**) immersion corrosion rate [30]; and (**c**) immersion H_2_ gas evolution [18].

**Figure 5 jfb-14-00242-f005:**
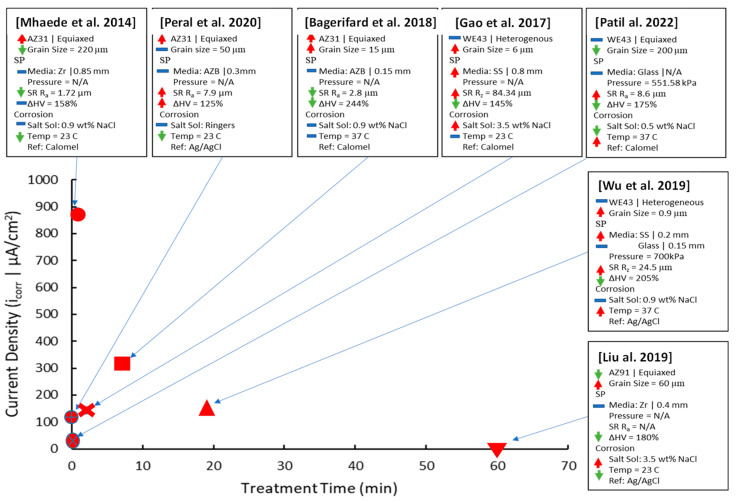
The corrosion density (i_corr_) of SP-treated samples against the treatment time parameter [5,18,28,29,30,48,50]. Arrows have been used to qualitatively represent the effects of each property on i_corr_, with lower (green), neutral (blue), and raised (red) arrows being used.

**Figure 6 jfb-14-00242-f006:**
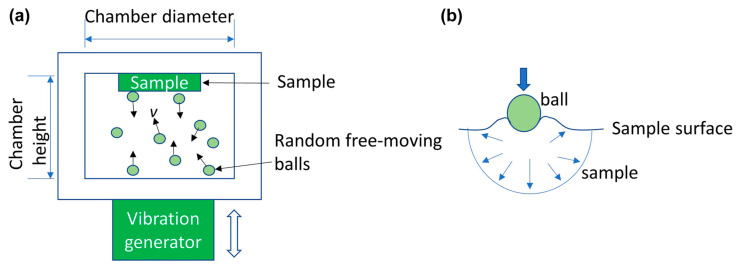
Illustration of surface mechanical attrition treatment [52] (**a**) its process and (**b**) deformation mechanics.

**Figure 7 jfb-14-00242-f007:**
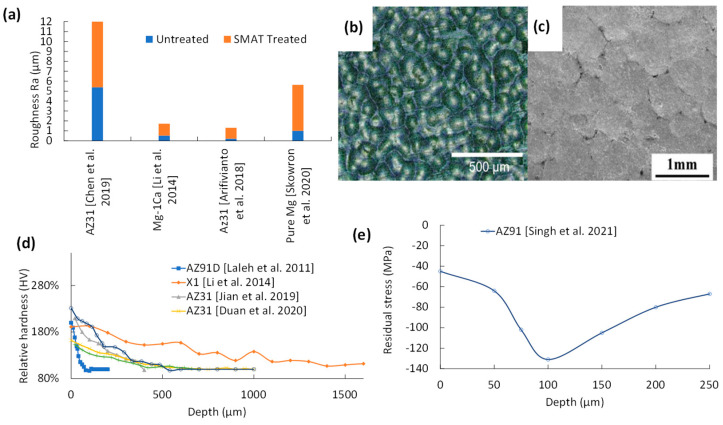
Effect of SMAT on (**a**) surface roughness [20,33,34,53], (**b**,**c**) surface topography due to media indents [31,35], (**d**) hardness [20,35,54,55] and (**e**) residual stress [56] of Mg alloys.

**Figure 8 jfb-14-00242-f008:**
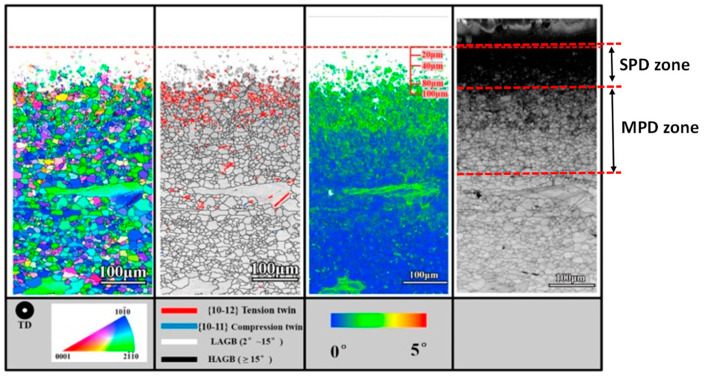
EBSD capture of microstructural change after SMAT treatment for AZ31 [57].

**Figure 9 jfb-14-00242-f009:**
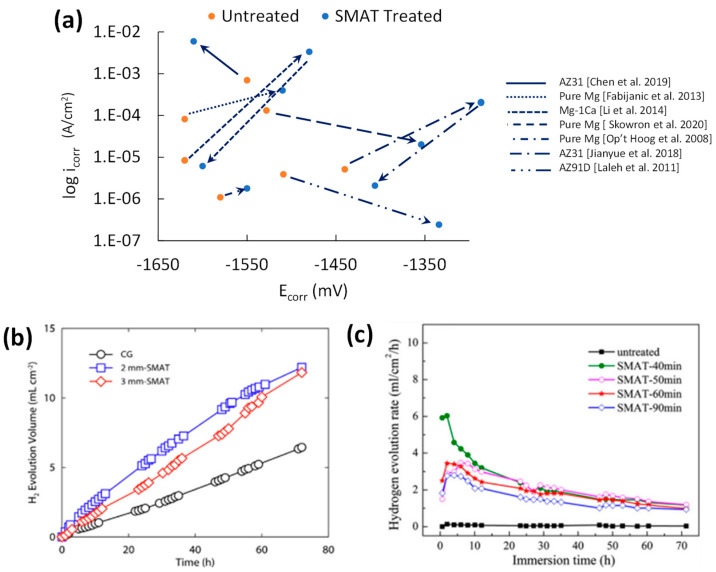
Effect of SMAT on (**a**) potentiodynamic corrosion resistance [20,32,33,34,35,54,58], (**b**) effect of SMAT media size on H_2_ evolution [34], and (**c**) effect of treatment time on H_2_ evolution [20].

**Figure 10 jfb-14-00242-f010:**
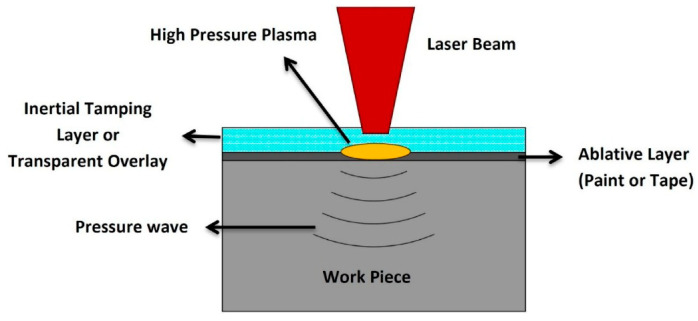
Schematic diagram of the laser shock peening process [59].

**Figure 11 jfb-14-00242-f011:**
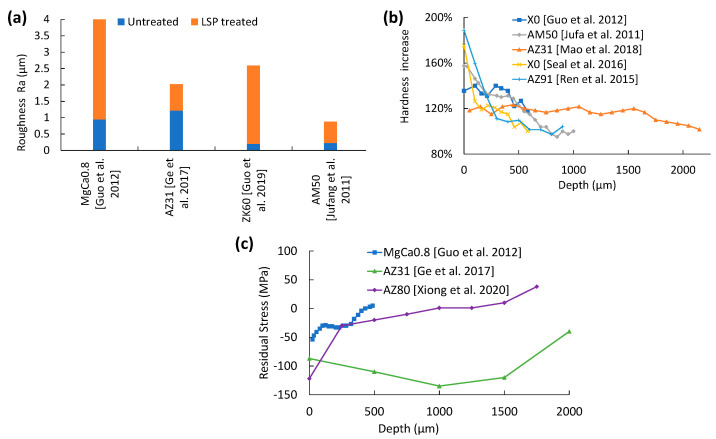
Effect of LSP on (**a**) surface roughness [19,36,37,39], (**b**) hardness [36,37,61,62,63], and (**c**) residual stress [19,36,38] for Mg alloys.

**Figure 12 jfb-14-00242-f012:**
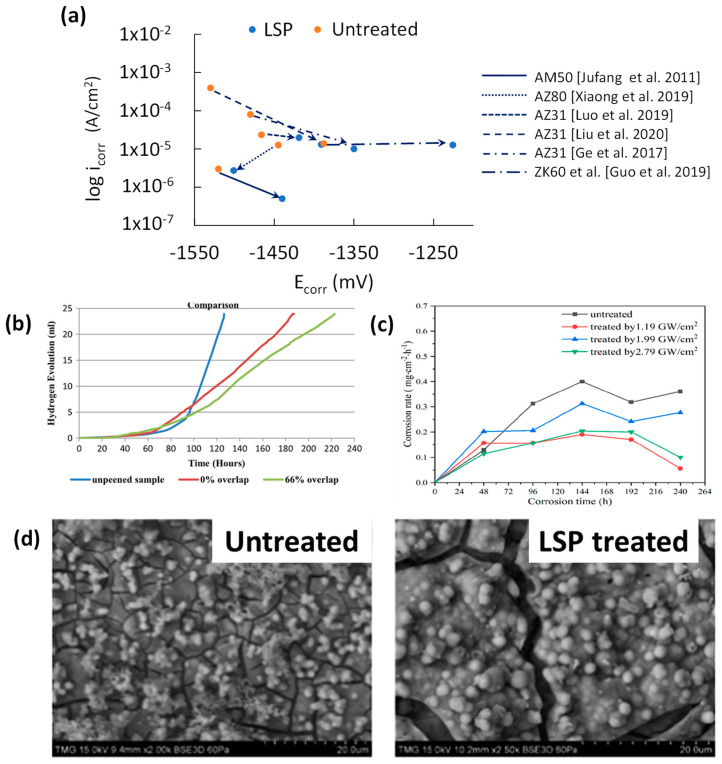
Effect of LSP on (**a**) potentiodynamic corrosion resistance [19,37,38,39,64,65], (**b**) effect of LSP overlap on H_2_ generation [66], (**c**) corrosion rate [39], and (**d**) corroded surfaces of untreated and LSP surfaces after 20 h of immersion [66].

**Figure 13 jfb-14-00242-f013:**
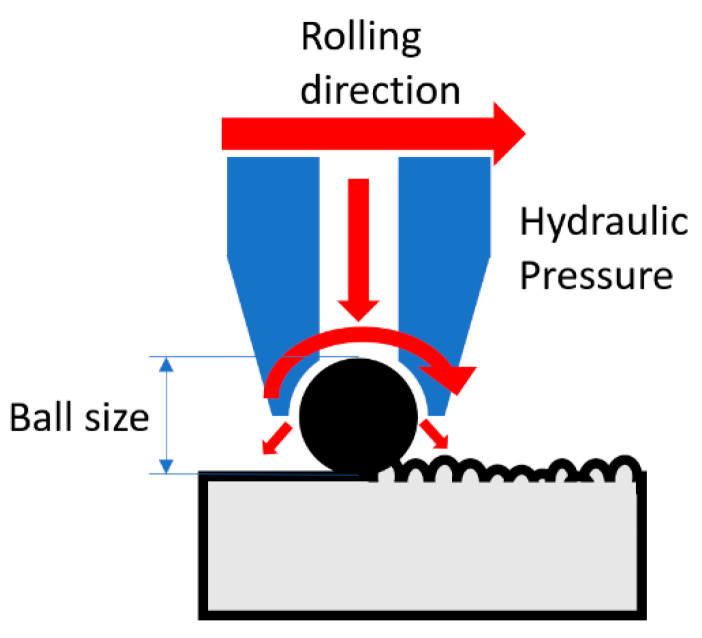
Illustration of the ball burnishing process.

**Figure 14 jfb-14-00242-f014:**
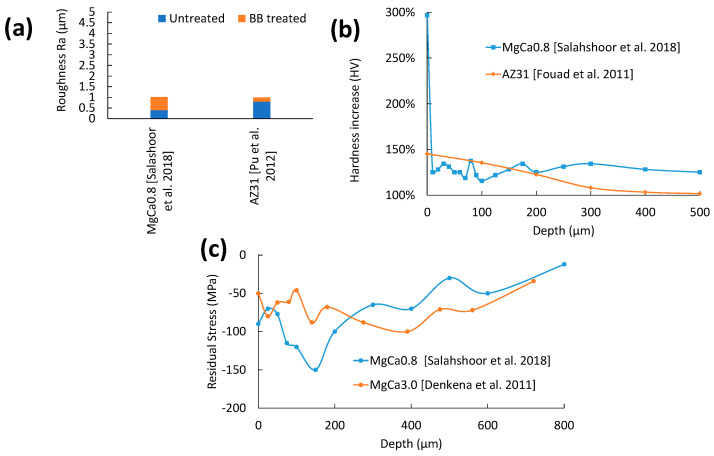
Effect of BB on (**a**) surface roughness [6,40], (**b**) hardness change [40,69], and (**c**) residual stress [16,40] of Mg alloys.

**Figure 15 jfb-14-00242-f015:**
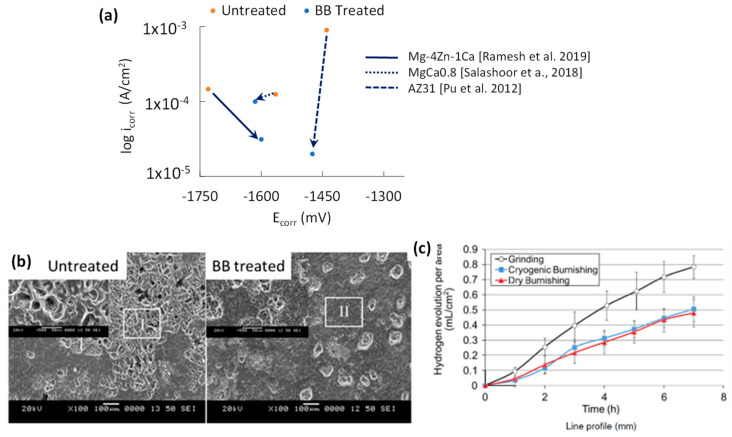
Effect of BB on (**a**) potentiodynamic corrosion resistance [6,17,40]; (**b**) SEM photos of corroded surfaces of untreated and BBed samples after immersion [17]; and (**c**) effect of cryogenic burnishing on H_2_ generation after 8 h of immersion [6].

**Figure 16 jfb-14-00242-f016:**
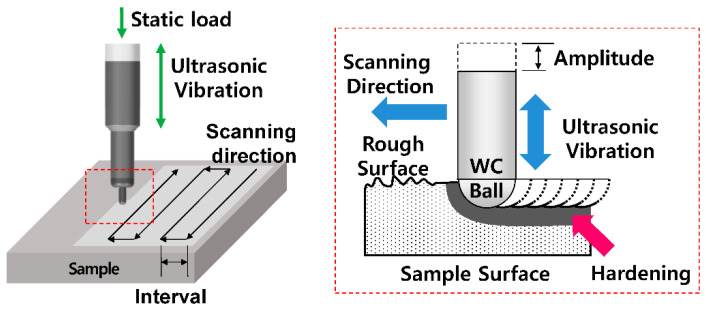
Schematic diagram of ultrasonic nanocrystal surface modification [71].

**Figure 17 jfb-14-00242-f017:**
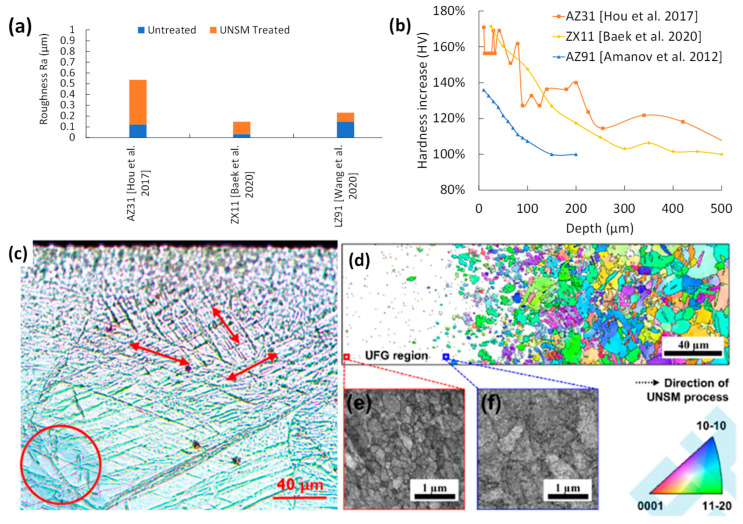
Effect of UNSM on (**a**) surface roughness [42,44,45], (**b**) hardness change [42,44,72], (**c**) grain refinement (red arrows indicate twinning formation due to UNSW while red circle the untreated grains of larger size) [42], and (**d**–**f**) microstructural change [44].

**Figure 18 jfb-14-00242-f018:**
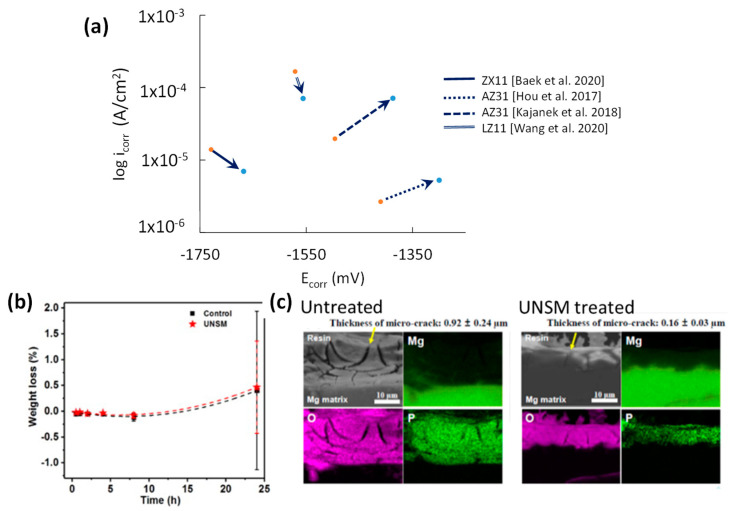
Effect of UNSM on (**a**) potentiodynamic corrosion resistance [42,43,44,45], (**b**) mass loss after 25 h of immersion [42], and (**c**) EDS spectra showing the elemental presence, cracks, and pitting of the corroded surface [44].

**Figure 19 jfb-14-00242-f019:**
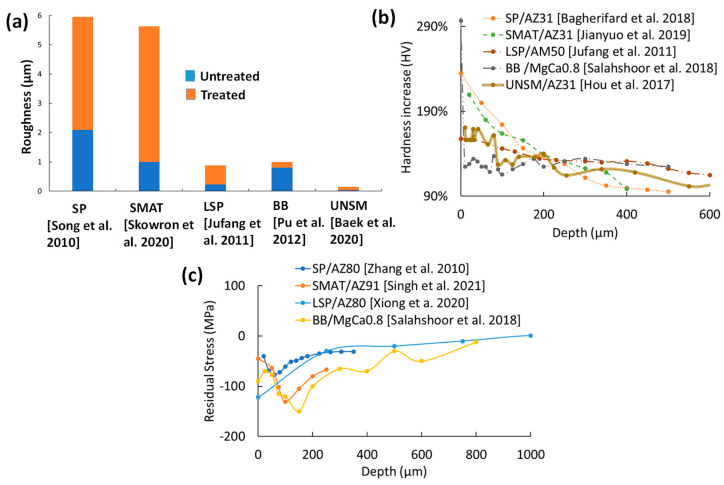
(**a**) Representative average roughness (Ra) for each treatment; SP [73], SMAT [33], LSP [37], BB [40], and UNSM [44], (**b**) relative hardness of each surface mechanical treatment [29,37,42,54,69], and (**c**) residual stress curves from all treatments except UNSM [30,38,40,56].

**Figure 20 jfb-14-00242-f020:**
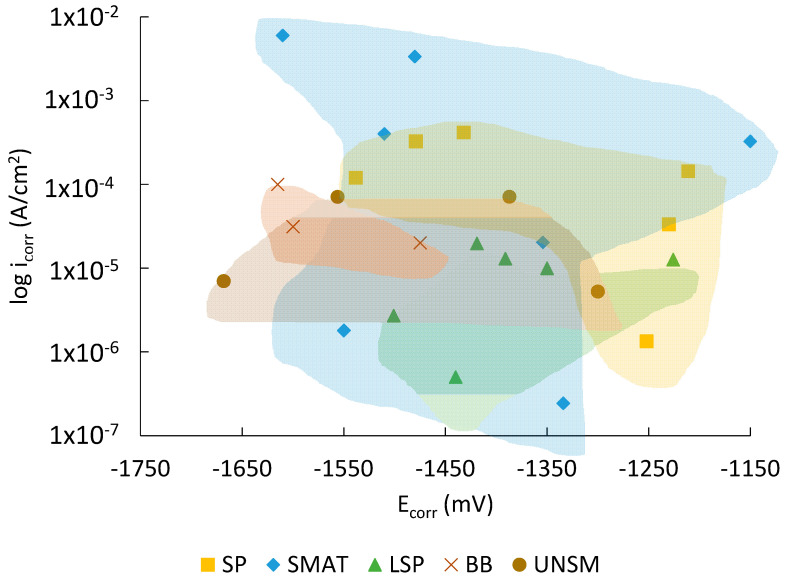
Every SP (■), SMAT (♦), LSP (▲), BB (✕) and UNSM (●) Tafel plot intercept excluding the untreated samples.

**Figure 21 jfb-14-00242-f021:**
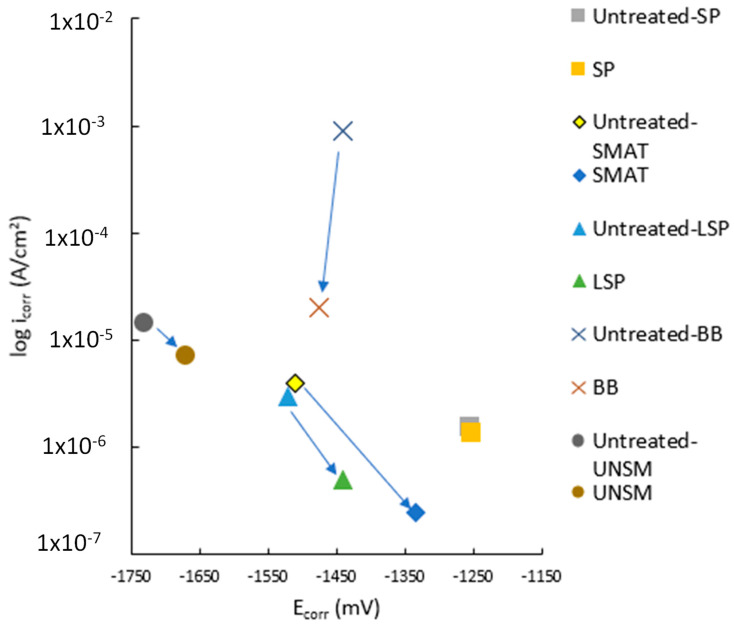
Simplified Tafel plot intercept with the lower icorr values per treatment type. SP (■) [18], SMAT (♦) [35], LSP (▲) [37], BB (✕) [6], and UNSM (●) [44].

**Figure 23 jfb-14-00242-f023:**
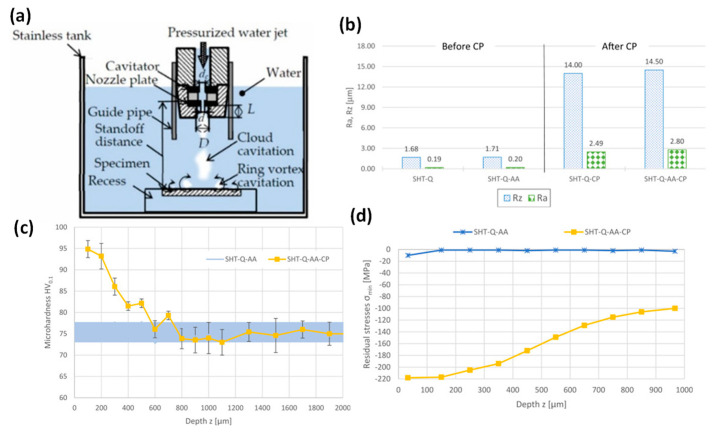
(**a**) Cavitation peening on AZ80A Mg alloys and comparison of (**b**) surface roughness, (**c**) residual stress, and (**d**) microhardness improvement [80].

**Figure 24 jfb-14-00242-f024:**
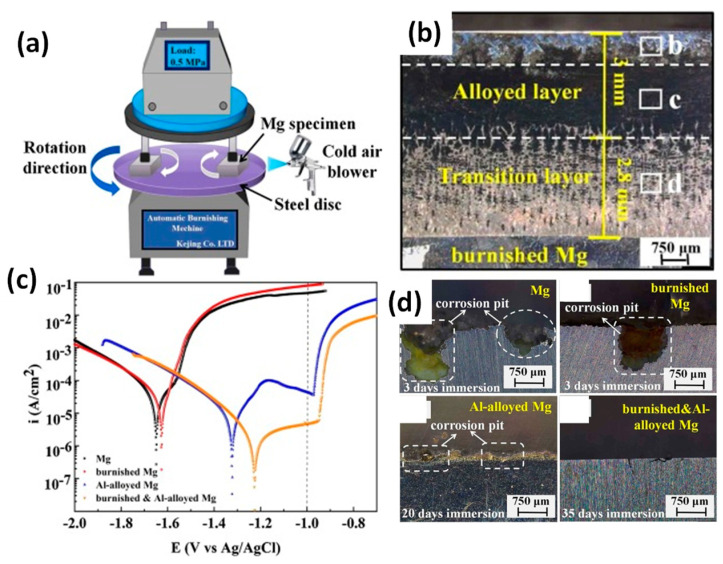
(**a**) Burnishing and Al alloying via thermal diffusion; (**b**) cross-sectional microstructure of burnished-Al alloyed surface; (**c**) Tafel plot showing corrosion characteristics; and (**d**) pitting corrosion micrographs [82].

**Figure 25 jfb-14-00242-f025:**
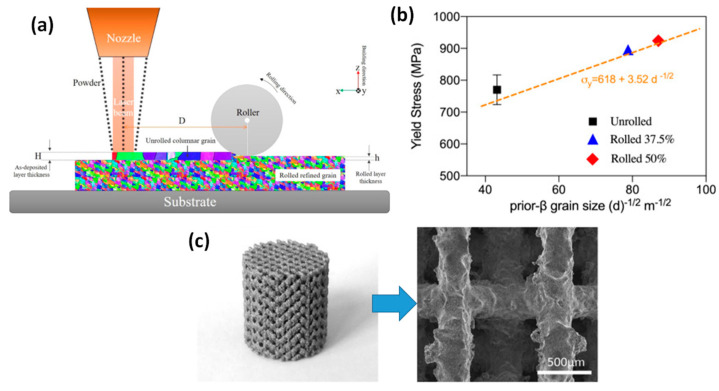
(**a**) In-situ roller burnishing in direct energy deposition (DED) of Ti6Al4V; (**b**) effect of grain refinement on strength improvement due to surface treatment [84]; and (**c**) as-printed WE43 Mg scaffold and surface morphology of an as-polished strut [85].

**Table 1 jfb-14-00242-t001:** Summary of surface integrity and corrosion results for Mg alloys treated by different surface modification processes.

Surface Treatment Strategy	Substrate Material	Bulk Material Condition	Surface Roughness		Microstructure	Microhardness		Residual Stress (RS)		Potentiodynamic Polarization Tests	
			Treated R_a_ (μm)	ΔR_a_ (μm)	Modified Layer Thickness (μm)	Hardness Increase (%)	Depth of Hardness Increase (μm)	Peak CRS (MPa)	Depth of CRS Change (μm)	Untreated/Treated E_corr_ (mV)	Untreated/Treated i_corr_ (μA/cm^2^)
SP	AZ31 [28]	Annealed	7.9	5.9	61	125	350	−50	145	−1509/−1538	29/120
	AZ31 [29]	Annealed	2.8	1.7	160	244	300	−65	550	−1542/−1479	21/326
	AZ91 [18]	As-cast	-	-	-	180	210	-	-	−1254/−1252	1.49/1.34
	WE43 [30]	Hot-rolled	24.49	16.88	180	205	1600	−250	500	−1414/−1211	208/143
	WE43 [31]	Rolled	8.6	6.3	-	175	-	−69	-	−1410/−1230	13.1/33.3
SMAT	Pure Mg [32]	As-cast	-	-	1000	-	-	−12.4	-	−1620/−1510	82/400
	Pure Mg [33]	Annealed	4.63	3.63	900	2	900	-	-	−1528/−1.354	132/20.3
	AZ31B [34]	Extruded	20.68	15.3	-	-	-	-	-	−1550/−1610	700/6000
	AZ91D [35]	Extruded	4	-	50	200	110	-	-	−1509/−1334	3.906/0.243
	Mg-1Ca [20]	As-cast	1.2	0.68	500	212	1350	-	-	−1620/−1480	8.45/3360
LSP	MgCa0.8 [36]	Extruded	3.5	2.55	-	144	600	−55	500	−1388/−1226	13.78/12.67
	AM50 [37]	As-cast	0.65	0.42	-	158	800	-	-	−1520/−1440	3/0.5
	AZ31B [19]	Wrought	0.807	−0.409	-	-	-	−136	2500	−1480/−1350	80/10
	AZ80 [38]	Hot-rolled	-	-	25	-	-	−122	1000	−1445/−1501	12.6/2.72
	ZK60 [39]	Extruded	2.39	2.19	-	-	-	−47.2	-	−1400/−650	13.78/12.67
BB	MgCa3.0 [16]	Extruded	0.9	−2.8	1100	-	-	−100	1000	-/-	-/-
	MgCa0.8 [40]	Annealed	0.62	0.22	-	244	100	−150	800	−1565/−1615	125/100
	AZ31B [6]	-	0.8	−0.22	3100	-	-	-	-	−1440/−1475	900/20
	AZ31 [41]	Annealed	-	-	940	-	-	-	-	-/-	-/-
	ZX41 [17]	Annealed	0.129	-	240	238	450	-	-	−1730/−1600	146.5/31.2
UNSM	AZ31B [42]	Annealed	0.414	0.292	400	164	550	-	-	−1410/−1300	2.67/5.27
	AZ31 [43]	Annealed	56	24	-	-	-	-	-	−1729/−1668	14/7
	ZX11 [44]	Annealed	0.114	0.081	200	-	-	-	-	−1496/−1387	19.7/71.3
	LZ91 [45]	Annealed	0.08	−0.07	-	170	600	-	-	−1571/−1556	166.5/70.8

**Table 2 jfb-14-00242-t002:** Reordered LSP data that connects ascending pulse energy density (PED) to i_corr_.

			Treated	
Substrate	Bulk Material Condition	Pulse Energy Density (GW/cm^2^)	E_corr_ (mV)	i_corr_ (μA/cm^2^)
AZ80 [38]	Hot-rolled	2.18	−1501	2.72
AM50 [37]	As-cast	3	−1440	0.5
ZK60 [39]	Extruded	5.1	−1226	12.67
AZ31B [19]	Wrought	14.8	−1350	10
AM50 [65]	As-cast	22.8	−1419	19.8
AZ31B [64]	Hot-rolled	27.2	−1391	13.04

**Table 3 jfb-14-00242-t003:** Reordered BB data that connects ascending roughness and grain size to i_corr_.

	SR	Microstructure	Electrochemical Data	
Substrate	Treated Ra (μm)	Treated Ave Grain Size (μm)	Treated E_corr_ (mV)	Treated i_corr_ (μA/cm^2^)
AZ31B [6]	0.2	1.4	−1475	20
ZX41 [17]	0.129	3.1	−1600	31.2
MgCa0.8 [40]	0.62		−1615	100

**Table 4 jfb-14-00242-t004:** Reordered UNSM data with ascending i_corr_.

		SR	Microhardness	Treated	
Substrate	Load (N)	Treated [Ra (μm)|Rz (μm)]	Peak Hardness	E_corr_ (mV)	i_corr_ (μA/cm^2^)
AZ31B [42]	5	0.41	95	−1300	5.27
ZX11 [44]	20	0.11	108	−1668	7
LZ91 [45]	1.4	0.08	85	−1556	70.8
AZ31 [43]	85	**56**		−1387	71.3

## Data Availability

Supporting data is not available in this article.

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
