# Peer review of "Mechanical Surface Treatments for Controlling Surface Integrity and Corrosion Resistance of Mg Alloy Implants: A Review"

_jfb, 2023, doi:10.3390/jfb14050242_

Round 1

Reviewer 1 Report

The paper provides summary of different processes very well. However, I feel the paper needs more critical analysis. Currently, there is only one summary table and no other critical analysis. There should be several analysis tables and figures in a review paper. The authors can only add their own contribution by those critical analysis tables and figures. Also, the future research section needs to be extended. I don't think the summary and outlook is good enough. There should be a separate section on future research directions, from which the future researchers can be benefitted. 

Author Response

Reviewer 1

Comments and Suggestions for Authors

The paper provides summary of different processes very well. However, I feel the paper needs more critical analysis. Currently, there is only one summary table and no other critical analysis. There should be several analysis tables and figures in a review paper. The authors can only add their own contribution by those critical analysis tables and figures.

Also, the future research section needs to be extended. I don't think the summary and outlook is good enough. There should be a separate section on future research directions, from which the future researchers can be benefitted. 

Reply: The authors thank the reviewer for the comment. We have included additional figures and tables with further analysis to explain the underlying implications of the results.

In particular, we have added a new section ‘9.0 Comparative analysis of surface treatments’ which critically and closely analyses the performance in terms of surface properties and corrosion resistance of SP, SMAT, LSP, BB and UNSM treatment strategies. Figures 20-22 are the new figures included as well to support the discussion. Changes are shown by blue-colored texts.

As suggested, we have also included a section named ‘Future research directions’ after the ‘Summary and outlook’ section in the revised manuscript. Changes are highlighted by blue-colored texts in the revised version.

Reviewer 2 Report

The submitted manuscript entitled ‘Mechanical surface treatments for controlling surface integrity and corrosion resistance of biodegradable Mg alloys’. This manuscript deals with the Microstructure evolution, mechanical and corrosion property of Mg alloy functional materials composite especially for orthopedic biomaterial.The manuscript is interesting, the main concern in this Reviewer is connected to the novelty of the investigations, such tests were performed in the past to the best knowledge of this Reviewer. Besides this problem a list of other technicalities arose.

1.     Title of the article is not suitable, need to rewrite it properly and make proper informative.

2.     Authors mentions the biological properties in the title but no any discussion into manuscript why?

3.     Introduction is very poor need to rewrite.

4.     Authors need to mention the biological discussion in the result and discussion section separately.

5.     The Conclusion section is too poor. Authors give the conclusions on every outcome.

6.     What is the novelty of this research?

Summarizing, I recommended the paper for publication with major revisions

Author Response

Reviewer 2

Comments and Suggestions for Authors

The submitted manuscript entitled ‘Mechanical surface treatments for controlling surface integrity and corrosion resistance of biodegradable Mg alloys’. This manuscript deals with the Microstructure evolution, mechanical and corrosion property of Mg alloy functional materials composite especially for orthopedic biomaterial. The manuscript is interesting, the main concern in this Reviewer is connected to the novelty of the investigations, and such tests were performed in the past to the best knowledge of this Reviewer. Besides this problem a list of other technicalities arose.

  1. Title of the article is not suitable, need to rewrite it properly and make proper informative.

Reply: The focus of the paper is the effect of different surface treatments on surface integrity and corrosion resistance of Mg alloys that can be used as implants. Biological properties of Mg alloys are however not the current scope of the paper. As suggested by the reviewer, we have slightly revised the title as “Mechanical surface treatments for controlling surface integrity and corrosion resistance of Mg alloys implants: a review” to reflect the focus and discussion of the current paper.  

  1. Authors mentions the biological properties in the title but no any discussion into manuscript why?

Reply: The title of the paper has been revised to reflect the scope and content of the paper.

  1. Introduction is very poor need to rewrite.

Reply: We have revised the ‘Introduction’ by further analysis of background, leading to a research gap which forms the rationale for conducting this review study. Changes and new additions are highlighted by blue-colored text within the revised manuscript.

  1. Authors need to mention the biological discussion in the result and discussion section separately.

Reply:

  1. The Conclusion section is too poor. Authors give the conclusions on every outcome.

Reply: As suggested by the reviewer, we have revised the Conclusions by summarizing the key insightful findings, along with an ‘Outlook’ and ‘Future research directions’ to better reflect the underlying discussion and focus of the paper. See the changes in blue-colored texts.  

  1. What is the novelty of this research?

 Reply: Surface treatments were studied a lot in literature, but in a broader perspective. Most, as-built materials are not usable, requiring some levels of surface treatments or post-processing to achieve the designed properties. The main contribution of our paper is to comprehensively review and carefully look into how the process mechanics (e.g. grain refinement, phase dissolution) and parameters (e.g. treatment intensity, media, time) affect the surface properties that control the corrosion resistance of Mg alloys. A comparative analysis along with challenges and prospects of the surface treatments is presented for readers to appreciate the benefits and issues. We have added further comments in the ‘Introduction’ section (see blue-colored texts) to clearly differentiate the novelty and originality of the work presented in this paper.

Summarizing, I recommended the paper for publication with major revisions.

Reviewer 3 Report

This paper reviews the current state-of-the-art mechanical 9 surface modification technologies and their response in terms of surface roughness, surface texture 10 and microstructural change due to cold work-hardening, affecting surface integrity and corrosion resistance of different Mg alloys. The summary is reliable and informative which is meaningful for other researchers. Thus, it is recommitted to be published. There are some minor revisions can be addressed before published.

1. For the title, it is better to pointed out the review type, like review: ..”so that this paper can be cited more.

2. Some figures can be more clearly, like Fig. 1, Fig.4, Fig.7, Fig.11 and so on.

3. Some opinions by authors can be added in the paper, especially in each part or technologies.  

Author Response

Reviewer 3

Comments and Suggestions for Authors

This paper reviews the current state-of-the-art mechanical 9 surface modification technologies and their response in terms of surface roughness, surface texture 10 and microstructural change due to cold work-hardening, affecting surface integrity and corrosion resistance of different Mg alloys. The summary is reliable and informative which is meaningful for other researchers. Thus, it is recommitted to be published. There are some minor revisions can be addressed before published.

  1. For the title, it is better to pointed out the review type, like review: ..”so that this paper can be cited more.

Reply: We have revised the title by included the word ‘a review’ in the end of the title as: “Mechanical surface treatments for controlling surface integrity and corrosion resistance of Mg alloys implants: a review

  1. Some figures can be more clearly, like Fig. 1, Fig.4, Fig.7, Fig.11 and so on.

Reply: As suggested by the reviewer, we have taken every endeavour to make all these figures more legible for readers to grasp the underlying fact and information.

  1. Some opinions by authors can be added in the paper, especially in each part or technologies.  

Reply: We have further commented on the results to provide an overview of significance of the results. We have added a new section ‘Comparative analysis of surface treatments’ and given opinions for readers to quickly digest the key messages of the review on different surface treatments for biodegradable Mg alloys.

Author Response

Reviewer 4

Comments:

The manuscript reports “Mechanical surface treatments for controlling surface integrity and corrosion resistance of biodegradable Mg alloys”. Considering the importance of Mechanical surface treatments, the review is timely. Overall, the paper is well-organized and the detailed discussion has made it easy to be understood. However, it requires minor revision before considering for publication.

  1. Page 3, What are the challenges posed by new material, geometry, manufacturing, and application, and why is a better understanding and evaluation of existing treatment techniques required?

Reply: New Mg alloys are being constantly developed with new alloying elements for improved corrosion resistance and biocompatibility. Also, different fabrication processes such as casting, additively manufacturing are used to manufacture these alloys in many geometric forms for implant applications. Regardless, the manufactured Mg alloys still suffer the fast corrosion because of its composition, microstructure. Different surface treatments are being employed to address this issue.

There is a good abundance of successful treatment strategies being used for enhancing surface, corrosion ad fatigue properties. However, this is still lack of studies that comprehensively focuses on comparative performance analysis from process mechanics point of view, i.e. how each process can influence microstructural change. By knowing this through the review will help researchers understand the capability of the current processes and explore new ways of treatment for even enhanced surface integrity and corrosion resistance for Mg alloys.

The current review paper takes on this initiative to give an insightful commentary for researchers on Mg surface treatment approaches – what is done so far, what can be more done and what can be explored to develop a successful Mg-based biodegradable implants.

  1. Can you please, discuss the key factors that play a crucial role in augmenting surface integrity and degradation characteristics of Mg alloys, and how do they interact with each other?

Reply: As explained in the revised manuscript, SP peening intensity and time are the key parameters affecting the grain refinement, residual stress and surface roughness. While grain refinement helps increase corrosion resistance, high surface roughness, tensile stress and cracks deteriorate corrosion resistance. SMAT results in similar surface topography of SP, but higher treatment time influenced significantly roughness and grain refinement, which improve corrosion resistance. For LSP, higher pulse energy density and coverage are two main parameters affecting surface roughness, stress and grain refinement.

Unlike SP, LSP does not create surface cracks, but shallow grain refinement depth. BB force is the dominant parameter affecting roughness and depth of modification. As compared to SP and LSP, BB smoothens surface significantly that results in higher corrosion resistance. UNSM uses vibration causing intervals of repeated forces that result in smoother surface. Vibration frequency, force, step and feedrate are the key parameters affecting underlying microstructural change and hardness. Vibratory motion with lower force increases the corrosion resistance.

  1. How does the alloy composition and concentration/type of saline medium affect the local and global corrosion of treated surfaces?

Reply: Yes, saline media and concentration can affect the corrosion performance. Higher concentrations can accelerate the corrosion due to higher dissolved ions. In most studies we reviewed, NaCl of 0.9-3.5% are used as corrosion medium when evaluating corrosion of Mg alloys. SBF and PBS medium were used in some studies as well. Corrosive media can play a key role in initiation and propagation of corrosion, resulting in different corrosion forms of Mg alloys. For example, NaCl causes filiform corrosion while Na2SO4 medium causes pitting type corrosion.

Note that the effect of corrosion media is not the scope of the current paper, and the details are not analysed. However, the corrosion media must be analysed when discussing the local and global corrosion mechanisms of Mg alloys. Thus we have included a comment on this in the ‘Future research directions’ section of the revised version (see the blue coloured texts).

  1. Which treatment techniques are able to increase surface finish, and which are not, and why?

Reply: LSP, BB and UNSM increase surface finish while SP and SMAT deteriorates the surface finish. In SP, hard particles with high kinetic energy randomly blasted onto surface causing significant indents, hence increasing surface roughness. On the other hand, BB or LSP or UNSM are controlled contact based process, causing plastic deformation, hence smoothening out the surface.  

  1. Which processing parameters affect the density and level of grain refinement induced by treatment techniques, and which techniques outperform the others in terms of grain refinement?

Reply: All treatment techniques are able to cause grain refinement within sub-surface at a certain depth, but the density and level of refinement is highly dependent on the key processing parameters including deformation intensity, media, overlap, time, pressure and load. Predominantly, SP, SMAT and LSP outperform over UNSM and BB in terms of deeper and finer grain refinement.

  1. What is the order of the treatment techniques in terms of microhardness magnitude and gradient, and how does it relate to compressive residual stress and depth?

Reply: In terms of magnitude and gradient, SMAT shows the highest microhardness followed by SP, LSP, UNSM and BB, in descending order. However, the compressive residual stress and depth induced by SP and SMAT are lower than that by either LSP or BB due to undesired high roughness and surface defects because of severe nature of plastic deformation.

  1. Which treatment techniques are more corrosion-resistant, and why? What are the key factors that contribute to increased corrosion resistance in these techniques?

Reply: Electrochemical corrosion results showed that LSP and BB treated surfaces were more corrosion resistant (i.e. lower icorr and higher Ecorr), compared to other techniques. The lower surface roughness, moderate nanograin refinement and lack of surface defects are responsible for the increased corrosion resistance. Similar trend was noted for long-term immersion in terms of H2 generation, mass loss, and pitting growth. 

  1. How do long-term immersion studies affect the corrosion resistance and other surface characteristics of Mg alloys treated with different surface treatment techniques?

Reply: Deeper and finer grains within surface induced by the surface treatment increase long term corrosion resistance such as anti-pitting. In this regard, SP and SMAT outperform over others. However, twining, surface cracks and pores can accelerate the corrosion.  Therefore, initial top surface integrity is critical for long term corrosion. BB, LSP or UNSM creates a good barrier layer at the initial stage thus preventing long term pitting. Hybrid surface treatment combining coating and mechanical treatment augment such corrosion performance.

  1. What are the potential applications of controlled surface plastic deformation via mechanical surface treatment techniques in the field of Mg alloy surface engineering, and what future research directions should be pursued in this area?

Reply: Potential application of this controlled surface plastic deformation is the fabrication of bone implants (e.g. bone plate, screws, cylinders). The process can also be used to improve fatigue life of implants and other industrial components such as gears, impellers, splines.

Reviewer 5 Report

In the presented Manuscript, authors summarized mechanical surface treatments and their effect on corrosion resistance of Mg alloys. Surface state of Mg alloys and its influence on corrosion is fundamental since Mg is highly reactive metal. Therefore, comprehensive review for these issue is vital. Neverthless, the Review contains essential informations there are some questions to the presented state: 

1.   Authors are suggested to enrich Introduction in the part where conversion coating is mentioned with the knowledge about surface treatmens such as Plasma electrolytic oxidation. As this process is applied in biomedicine and is considered to be effective way towards reducing corrosion rate.  Suggested reference is recommended as follows : 

Štrbák, M., Kajánek, D., Knap, V., Florková, Z., Pastorková, J., Hadzima, B., & Goraus, M. (2022). Effect of plasma electrolytic oxidation on the short-term corrosion behaviour of AZ91 magnesium alloy in aggressive chloride environment. Coatings12(5), 566.

2. Lines (39-40). It is recommended for authors to add informations what consequencies causes higher corrosion of magnesium in human body. Recommendation is oriented on the undesirable hydrogen evolution and the belonging issues as a consequence of the electro-chemical reaction.  

3. Lines (114-115). Here is missing refference) (Error Ref- 114 erence source not found.) 

4. Lines (122-123). It is recommended to add more informations why the higher Ecorr value is desirable.

5. Lines (192). Authors are suggested to check whether β-phase in AZ91 has different chemical formula than the one written manuscript (Al17Mn14) because β-phase in AZ91 should have different chemical composition based on the literature. 

6.  Explanation why galvanic cell is important in case of Mg and its alloys. Some informations about cathode and anode should be added. As it is one  of the main issues why magnesium corrodes at micro-galvanic scale. 

7. Lines (266-280). Authors defined Surface roughness as SR. However, in lines (266-280) is the SR interchanged with RS. It is recommended to use uniform abbreviation in whole Manuscript. 

8. In the section Hybrid surface treatments 

Authors are suggested to add some results from other studies where mechanical surface was used as a pretreatment and where its impact was beneficial for resulting corrosion resistance. 

For instance in recent study authors investigated the effect of Shoot peening on the formation of PEO (MAO) coating : 

Kajánek, D., Pastorek, F., Hadzima, B., Bagherifard, S., Jambor, M., Belány, P., & Minárik, P. (2022). Impact of shot peening on corrosion performance of AZ31 magnesium alloy coated by PEO: Comparison with conventional surface pre-treatments. Surface and Coatings Technology446, 128773. 

9. Authors used for comparison in the graphs various Mg alloys including (AZ91, AZ31, WE43 and others). Explanation how certain elements influence the corrosion of magnesium alloys. Informations whether they are increasing or decreasing corrosion resistance could be added. As the alloying elements also influence PDP curves. 

Author Response

Reviewer 5

Comments and Suggestions for Authors

In the presented Manuscript, authors summarized mechanical surface treatments and their effect on corrosion resistance of Mg alloys. Surface state of Mg alloys and its influence on corrosion is fundamental since Mg is highly reactive metal. Therefore, comprehensive review for these issue is vital. Neverthless, the Review contains essential informations there are some questions to the presented state: 

1.   Authors are suggested to enrich Introduction in the part where conversion coating is mentioned with the knowledge about surface treatmens such as Plasma electrolytic oxidation. As this process is applied in biomedicine and is considered to be effective way towards reducing corrosion rate.  Suggested reference is recommended as follows : 

Štrbák, M., Kajánek, D., Knap, V., Florková, Z., Pastorková, J., Hadzima, B., & Goraus, M. (2022). Effect of plasma electrolytic oxidation on the short-term corrosion behaviour of AZ91 magnesium alloy in aggressive chloride environment. Coatings12(5), 566.

Reply: We have included the suggested reference along with a comment in 4th paragraph of Introduction section of the revised manuscript. Please see the blue-coloured text.

  1. Lines (39-40). It is recommended for authors to add informations what consequencies causes higher corrosion of magnesium in human body. Recommendation is oriented on the undesirable hydrogen evolution and the belonging issues as a consequence of the electro-chemical reaction.  

Reply: Higher Mg corrosion causes higher H2 gas generation. Human tissues around the implant site can handle a certain amount of H2 gas. However, excessive H2 gas accumulates in tissue cavities, reducing its ability to efficiently exchange and infiltrate gases for normal biological operations. As a consequence, excessive H2 gas cavities causes prolonged discomfort and disturbs the balance of blood cell parameters, thus decreasing the survivability.

We have added a comment on the consequence of H2 gas due to Mg corrosion in the 2nd paragraph of the Introduction of the revised manuscript.

  1. Lines (114-115). Here is missing refference) (Error Ref- 114 erence source not found.) 

Reply: We have revised and fixed the reference.

4. Lines (122-123). It is recommended to add more informations why the higher Ecorr value is desirable.

Reply: Higher Ecorr means the material is more noble and corrosion resistant. In other words, the system will rather take up electrons than loos electrons, so a reduction reaction is more likely, meaning less material degradation. Surface treatment aims to thus increase Ecorr. However, the corrosion current and rate may not be directly proportional to corrosion potential.

  1. Lines (192). Authors are suggested to check whether β-phase in AZ91 has different chemical formula than the one written manuscript (Al17Mn14) because β-phase in AZ91 should have different chemical composition based on the literature. 

Reply: We have revised the beta phase of AZ91 and written as Mg17Al12.

  1. Explanation why galvanic cell is important in case of Mg and its alloys. Some informations about cathode and anode should be added. As it is one  of the main issues why magnesium corrodes at micro-galvanic scale. 

Reply: Galvanic cell is a very popular and primary technique to determine the local corrosion characteristics, potential and pitting of a material. The approach is a very standard one. It is easy, simple and affordable to quickly assess material’s primary corrosion resistance. Many literature detailed the working principle of the process. Therefore, the authors did not explain much on the process mechanics, as they are very generic. In particular, we are interested in more Ecorr and Icorr of the treated Mg alloys which are two parameters that dictate the corrosion performance. Higher Ecorr and lower Icorr means higher corrosion resistance or lower degradation rate. Cathodic and anodic reaction curves intersect is a measure of Ecorr and Icorr of the material of interest.

  1. Lines (266-280). Authors defined Surface roughness as SR. However, in lines (266-280) is the SR interchanged with RS. It is recommended to use uniform abbreviation in whole Manuscript. 

Reply: We have changed RS to CRS (compressive residual stress) while SR only means Surface Roughness for consistency throughout the manuscript.

  1. In the section „Hybrid surface treatments “

Authors are suggested to add some results from other studies where mechanical surface was used as a pretreatment and where its impact was beneficial for resulting corrosion resistance. 

For instance in recent study authors investigated the effect of Shoot peening on the formation of PEO (MAO) coating : 

Kajánek, D., Pastorek, F., Hadzima, B., Bagherifard, S., Jambor, M., Belány, P., & Minárik, P. (2022). Impact of shot peening on corrosion performance of AZ31 magnesium alloy coated by PEO: Comparison with conventional surface pre-treatments. Surface and Coatings Technology446, 128773.

Reply: We have added the reference with a comment in Section 10 ‘Hybrid Surface Treatment’ as suggested by the reviewer. Please see the blue-coloured text within the revised manuscript.

  1. Authors used for comparison in the graphs various Mg alloys including (AZ91, AZ31, WE43 and others). Explanation how certain elements influence the corrosion of magnesium alloys. Informations whether they are increasing or decreasing corrosion resistance could be added. As the alloying elements also influence PDP curves. 

Reply: We have included and explained in details the Table 4, Table 5 and Table 6 for comparative PDP corrosion performance between different Mg alloys e.g. AZ91, AZ31, WE43 and others in Sections 6.2, 7.2 and 8.2 of the revised manuscript. Please see the blue-coloured texts within the sections.

Round 2

Reviewer 2 Report

Authors have done significant improvements in the revised manuscript. Recommended to acceptance.